# OceanVerse: Evaluable 4D Ocean Element Reconstruction Dataset under Realistic Sparsity

## Abstract

The vast oceans record the impacts of climate change and human activities on the Earth system. Over the past century, oceanographic scientists have collected extensive ocean profile data to reflect variations of oceanic elements, such as dissolved oxygen. However, due to the sophisticated measurements and high costs, historical ocean element observation data remains highly sparse and uneven across the global ocean, with the annual missing rate exceeding 90%. Thus, quantitatively understanding the four-dimensional (4D) spatiotemporal evolution of oceanic elements continues to pose a significant challenge. Machine learning (ML) techniques demonstrate superior capabilities in perceiving spatiotemporal variations within large-scale data, presenting promising opportunities to harness implicit correlations for global reconstruction. However, fragmented data and interdisciplinary differences create barriers to the availability of AI-ready open data, further hindering ML practitioners from designing specialized models. To solve this problem, we present the first oceanic 4D sparse observation reconstruction dataset, named OceanVerse. By integrating nearly 2 million real-world profiles since 1900 and three differentiated Earth system numerical simulation, we construct a comprehensively evaluable dataset with missing patterns that align with real-world conditions through a digital twin sampling. OceanVerse provides a novel large-scale ($\sim 100\times$ nodes vs. existing datasets) dataset that meets the MNAR (Missing Not at Random) condition, supporting more effective model comparison, generalization evaluation and potential advancement of scientific reconstruction architectures. The OceanVerse dataset and codebase are publicly available[1].

## 1 Introduction

71% of the Earth is covered by oceans. With the intensifying influence of human activities and global climate warming, the marine ecosystem is undergoing sustained and profound changes. While satellite remote sensing has significantly enhanced the data volume on physical variables of the *ocean surface* (Dohan, 2017; Morrow et al., 2019), such as sea surface height (SSH), sea surface temperature (SST), and ocean color, it is not feasible to measure the four-dimensional (4D) spatiotemporal distribution of underwater biogeochemical elements, including horizontal, vertical, and temporal variations. Instead, such measurements rely on vertical **ocean profiles** (Figure 1) collected through hydrographic surveys (Anderson, 2020), buoys (e.g., Argo (Jayne et al., 2017)), CTD sensors at various depths, which are costly and inefficent. Thus, the historical ocean profile data is highly uneven and sparse. For example, over the past 100 years, the observed dissolved oxygen accounts for only 3.735% of the entire ocean (Lu et al., 2024). Therefore, in order to reveal the impacts of climate change and human activities on marine ecosystems (Cheng et al., 2019; Breitburg et al., 2018), it is of great significance to accurately reconstruct the complete global ocean biogeochemical cycles (Moore et al., 2018; Visbeck, 2018).

AI-based spatiotemporal imputation (Liu et al., 2023a; Nie et al., 2024), as an promising approach, can reconstruct missing values based on sparse oceanic observation data. However, due to the disparity in disciplinary backgrounds, **the lack of public AI-ready oceanic datasets** poses a significant challenge for AI researchers to develop specific algorithms. Meanwhile, the insufficient test samples renders different AI methods difficult to evaluate quantitatively. Consequently, even the latest oceanography

---

[1]The OceanVerse resource is available at https://anonymous.4open.science/r/OceanVerse/.

studies are lagging behind advanced techniques and rely on spatial interpolation (Zhou et al., 2022) or well-established machine learning methods, such as random forests (Sharp et al., 2022) and shallow feedforward neural networks (Zhong et al., 2025). To bridge this gap, here we introduce the OCEANVERSE dataset using the widely recognized issue of the ocean oxygen cycle, which raises the following question:

*How can the* OCEANVERSE *dataset be designed to validate the fully reconstructed results and construct masks that satisfy the true missing patterns of the ocean?*

Existing data imputation benchmarks (Li et al., 2018; Yi et al., 2016; Zhu et al., 2024) are typically complete datasets, with missing values simulated by randomly generating missing masks. Since the data is complete, performance validation can be directly conducted on the fully reconstructed results. However, real-world ocean observations are inherently incomplete, and areas that were never observed in the past will never possess known true values. This makes it *impossible* to validate the fully reconstructed results. As a compromise, observed data can be partially masked to create test samples for evaluating imputation performance. Nevertheless, this masking approach does not align with the true missing patterns. Additionally, random masking can introduce spatiotemporal correlations between the training and test data, which negatively impacts performance evaluation and further exacerbates data sparsity.

To address this issue, OCEANVERSE proposes an innovative approach by constructing datasets through a digital twin model as shown in Figure 2. We propose a virtual Earth, parallel to the real Earth, where the dynamics of the system are globally known. Building on human historical observation on the real Earth (red dots), we sample data on the virtual Earth to obtain corresponding sampling observation (blue dots). The remaining data are considered missing values. Hereby, this allows us to perform a complete reconstruction evaluation, with the missing patterns aligning with the real-world missing patterns. Based on the OCEAN-VERSE dataset, we can train and comprehensively evaluate different AI models to select the most suitable model architecture for application on the real Earth. When we expand to more virtual Earths, we can further validate the model's generalization performance. We expect OCEAN-VERSE to facilitate new opportunities for two main communities. For AI for Science researchers, it offers a scientifically grounded ocean science benchmark for evaluating reconstruction skill under realistic sparse element observations. For ML method developers, it provides a standardized and challenging testbed for fair comparison of reconstruction model architectures. Overall, OCEAN-VERSE creates new opportunities for advancing spatiotemporal representation learning under higher data sparsity, larger-scale spatiotemporal scales, and more complex underlying mechanisms.

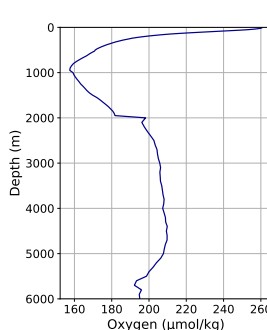

Figure 1: Historical average dissolved oxygen profile in January.

## 2 RELATED WORK

### 2.1 DATA-DRIVEN OCEAN SCIENCE DISCOVERY

In recent years, trained on meteorological reanalysis data (such as ERA5 (Hersbach et al., 2020)), AI models have achieved superior performance surpassing the traditional numerical weather forecasting, leading to a series of research advancements, including Pangu-Weather (Bi et al., 2023), ClimaX (Nguyen et al., 2023), GraphCast (Lam et al., 2023), and NeuralGCM (Kochkov et al., 2024). In contrast, breakthrough advancements have yet to emerge in the field of data-driven ocean science discovery, partly due to the lack of AI-ready ocean data. The domain expertise of ocean science erect a barrier to the development of AI technologies.

To address this issue, OceanBench (Johnson et al., 2023) pioneeringly provides AI-ready sea surface height data, obtained through satellite remote sensing, to support the ML-driven SSH mapping task. To further obtain the underwater data, remote sensing techniques are no longer applicable, and it becomes necessary to rely on profile data collected by research vessels, ocean buoys, submersibles, and ocean observation networks (Chen et al., 2022). However, these data are highly dispersed, which has led to the establishment of international research initiatives such as Argo (Jayne et al., 2017) and GEOTRACES (Anderson, 2020), aimed at promoting the integration and utilization of global ocean

observation data through cross-national collaboration and data sharing. Our work introduces the first AI-ready ocean profile dataset. We gather a substantial amount of historical ocean observation data from multiple sources. Additionally, through the use of digital twin sampling, we present three sets of Earth simulation results, aiming to advance data-driven understanding of the global deep ocean element cycling processes.

## 2.2 SPATIOTEMPORAL IMPUTATION DATASET

Existing spatiotemporal imputation datasets are mostly constructed by masking missing values based on complete datasets, e.g., Traffic speed dataset (PEMS-BAY (Li et al., 2018), META-LA (Li et al., 2018)), Traffic volume dataset (PEMS03, PEMS04, PEMS07, PEMS08 (Chen et al., 2001)), Air pollutant dataset (AQI, AQI-36 (Yi et al., 2016)), Energy power dataset (SO-LAR (Nie et al., 2024), CER-EN (Cini et al., 2022)), for effective performance evalution. Table 1 provides a detailed comparison among several popular spatio-temporal data imputation datasets and our OCEANVERSE dataset.

Rubin et al. (Rubin, 1976) classify missing data problems into three categories, i.e., missing completely at random (MCAR), missing at random (MAR), and missing not at random (MNAR). However, due to the lack of domain knowledge, most existing studies (Zhu et al., 2023; Nie et al., 2024; Cao et al., 2018; Tashiro et al., 2021) primarily construct missing data from the complete

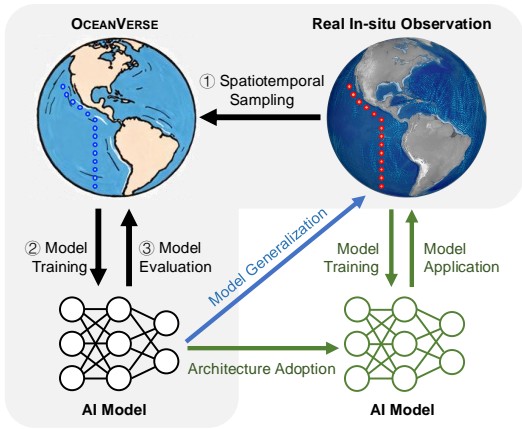

Figure 2: An Overview of the Construction and Application Process of the OCEANVERSE Dataset.

observations using MCAR and MAR settings. Under this artificially constructed missing data assumption, real-world missing patterns cannot be accurately reflected, as manually induced missingness tends to exhibit uniformity and temporal invariance, with spatial randomness. In contrast, real-world missing data patterns are shaped by factors such as data collection methods, transmission modes, and other contextual variables, leading to more complex and dynamic patterns in both time and space. A recent survey (Miao et al., 2022) indicates that nearly all algorithms perform worst under the MNAR scenario. However, due to the lack of sufficient MNAR datasets, it has been challenging to thoroughly validate and compare different baselines. In addition, existing datasets have limited spatial coverage, with small graph sizes (typically only a few hundred nodes), making it challenging to validate them in large-scale, complex interaction scenarios.

In this paper, we propose a large-scale spatiotemporal dataset OCEANVERSE, which utilizes the concept of digital twins to construct observations and missing data through real-world sampling points simulated on a virtual Earth. This approach truly reflects historical human exploration of the oceans, while the completeness of the simulated Earth data provides a ground truth for performance validation. Meanwhile, OCEANVERSE represents a significant contribution to filling the gap in existing datasets for evaluating the performance of various data reconstruction algorithms on large-scale (approximately 1 million nodes) and dynamic networks.

## 3 THE OCEANVERSE DATASET

### 3.1 PROBLEM FORMULATION

Sparse scientific observation reconstruction is a widely encountered problem in natural science. Particularly in oceanography, where observations of various elements in the ocean are often localized and short-term, the goal of scientific reconstruction is to piece together the data to derive a global spatiotemporal distribution.

**Task Definition**. Figure 3 illustrates the workflow of sparse scientific observation reconstruction. In this study, we use the reconstruction of dissolved oxygen as a case study to formally define

Table 1: Statistical comparison of public spatiotemporal imputation datasets and OCEANVERSE

| Dataset | #Nodes | #Edges | Spatial Corr. | Temporal Range | Interval | Missing |
|---|---|---|---|---|---|---|
| PEMS-BAY (Li et al., 2018) | 325 | 2,369 | Static | Jan 2017 - Jun 2017 | 5 min | Artificial |
| METR-LA (Li et al., 2018) | 207 | 1,515 | Static | Mar 2012 - Jun 2012 | 5 min | Artificial |
| PEMS03 (Chen et al., 2001) | 358 | 546 | Static | Sep 2018 - Nov 2018 | 5 min | Artificial |
| PEMS04 (Chen et al., 2001) | 307 | 338 | Static | Jan 2018 - Feb 2018 | 5 min | Artificial |
| PEMS07 (Chen et al., 2001) | 883 | 865 | Static | May 2017 - Aug 2017 | 5 min | Artificial |
| PEMS08 (Chen et al., 2001) | 170 | 276 | Static | Jul 2016 - Aug 2016 | 5 min | Artificial |
| AQI (Yi et al., 2016) | 437 | 2,699 | Static | May 2014 - Apr 2015 | 60 min | Artificial |
| AQI-36 (Yi et al., 2016) | 36 | 654 | Static | May 2014 - Apr 2015 | 60 min | Artificial |
| SOLAR (Nie et al., 2024) | 137 | 9,316 | Static | 2006 | 10 min | Artificial |
| CER-EN (Cini et al., 2022) | 485 | 4,365 | Static | 2016 | 30 min | Artificial |
| OCEANVERSE (`CESM2-omip1`) | 42,491 | 577,067 - 1,397,186 | Dynamic | 1948 - 2009 | 1 year | Real-world |
| OCEANVERSE (`CESM2-omip2`) | 42,491 | 834,100 - 1,397,186 | Dynamic | 1958 - 2018 | 1 year | Real-world |
| OCEANVERSE (`GFDL-ESM4`) | 42,491 | 395,616 - 1,397,199 | Dynamic | 1920 - 2014 | 1 year | Real-world |

the problem. Dissolved oxygen (DO) is a key variable that reflects both marine ecosystem health and climate change. Under ongoing anthropogenic impacts and global warming, the ocean is experiencing a pronounced deoxygenation crisis. However, due to the extreme sparsity of historical DO observations, large uncertainties remain regarding global deoxygenation trends, variability, and driving mechanisms. Let $X_{\text{obs}} \in \mathbb{R}^{N \times T \times D}$ represent the incomplete observation profiles of the target variable, where $N$ denotes the number of nodes, $T$ the number of timesteps, and $D$ the number of depth levels. Let $\Omega_{\text{obs}} = (\omega_{n,t,d})_{n,t,d} \in \{0,1\}^{N \times T \times D}$ be a binary indicator matrix representing the observed entries, where $\omega_{n,t,d} = 1$ if the entry at position $(n, t, d)$ is observed, and $\omega_{n,t,d} = 0$ if it is missing. Consequently, the missing observations can be represented as:

$$X_{\text{obs}} = \tilde{X}_{\text{obs}} \odot \Omega_{\text{obs}} + \emptyset \odot \left( \mathbb{1}_{N \times T \times D} - \Omega_{\text{obs}} \right),$$

where $\tilde{X}_{\text{obs}}$ denotes the complete ground truth, $\emptyset$ denotes the indicator of not available data observation, $\odot$ is the element-wise product and $\mathbb{1}_{N \times T \times D}$ is an $N \times T \times D$ matrix filled with ones.

Furthermore, let $X_{\text{env}} \in \mathbb{R}^{N \times T \times D \times K}$ denote the corresponding $K$ environmental variables. It is worth noting that due to differences in observational instruments and methods, the missing patterns of the target variable $\Omega_{\text{obs}}$ and different environmental variables $\Omega_{\text{env}}^k$ ($k = \{1, \cdots, K\}$) are distinct.

The goal of sparse scientific observation reconstruction aims to design a model $f(\cdot; \theta)$ parameterized with $\theta$ perform regression of the global target variable based on sparse observations of the target variable and other environmental variables: $f(X_{\text{obs}}, X_{\text{env}}; \theta) \rightarrow \tilde{X}_{\text{obs}}$.

**Task Comparison**. Sparse scientific observation reconstruction differs significantly from existing tasks. ❶ *Spatiotemporal imputation* (Cini et al., 2022; Nie et al., 2024; Yang et al., 2025) focuses on reconstructing the incomplete target variable by leveraging its internal spatiotemporal correlations, but does not incorporate the incomplete environmental variables which provide important context. ❷ *Multivariate time series imputation* (Cao et al., 2018; Tashiro et al., 2021; Drouin et al., 2022) tackles missing values by using cross-variable correlations between the target and environmental variables at a single spatial point. Yet, it overlooks the global spatial dependencies, which are crucial for understanding broader trends. ❸ *Spatiotemporal prediction* models (Lu et al., 2020; Zhang et al., 2020; Cini et al., 2023) aim to capture complex correlations in complete datasets, but they struggle to identify the patterns of missing data and the relationships between missing and observed samples, especially when data is highly sparse or entirely absent within a given time window. This gap in existing methods highlights the need for more robust techniques that can handle the complexities of sparse observational data across both time and space, incorporating all available variables and accounting for both local and global dependencies. Such advancements are essential for improving our understanding and management of complex environmental systems.

**Scientific Significance**. Sparse 4D observational data reconstruction helps to reveal the spatiotemporal evolution patterns of marine systems under the influences of climate change and human activities. For instance, the proposed OCEANVERSE dataset constructs data with ocean dissolved oxygen as the target variable, aiming to compare the quantitative analysis of ocean hypoxia through different AI methods. Ocean hypoxia (Schmidtko et al., 2017; Gong et al., 2021) refers to the phenomenon where the concentration of dissolved oxygen in seawater drops below the levels required for the survival of marine organisms. This typically occurs in deep-sea or nearshore areas (Breitburg et al.,

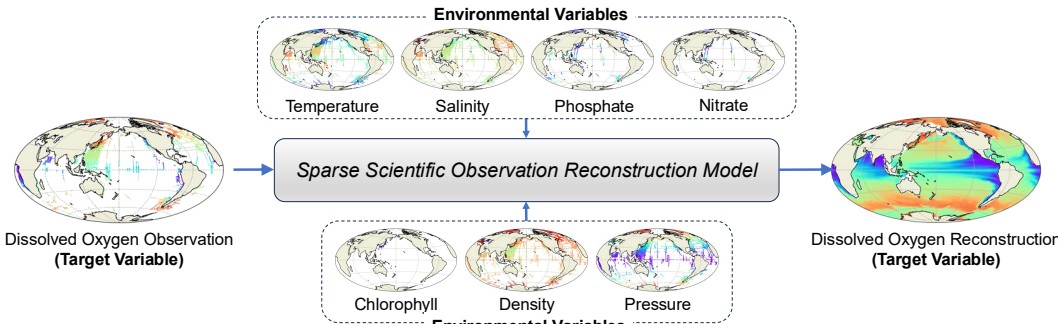

Figure 3: The task of sparse scientific observation reconstruction aims to perform regression of the global target variable based on sparse observations of the target variable (dissolved oxygen) and other environmental variables (temperature, salinity, phosphate, nitrate, chlorophyll, density, pressure, etc.). The inputs and outputs of the model constitute the OCEANVERSE dataset proposed in this paper.

2018; Stramma et al., 2008; Li et al., 2024), especially in regions affected by climate change and human activities such as overfishing, agricultural pollution, etc. However, due to the limitations of marine observation methods, obtaining high-quality data across the entire spatial and temporal range is very challenging. By applying sparse data reconstruction techniques, it is possible to recover the distribution of dissolved oxygen across the full range from limited observational data, which is crucial for revealing the health status of marine ecosystems and formulating conservation policy.

**Evaluation Requirements**. The evaluation of scientific data reconstruction requires a thorough and complete comparison to obtain reliable results. Previous extensive research (Lu et al., 2024; Sharp et al., 2022; Zhong et al., 2025) on the evaluation of scientific reconstructions has relied on *partial comparison*, where some observation data is masked, and the performance evaluation of these partial observations is deemed as the performance of the global-scale reconstruction. Since we can never obtain the ground truth of global historical values, this evaluation approach is indeed an acceptable but unsatisfied measure. However, we must acknowledge that this evaluation method introduces bias in the selection of the reconstruction model. To address this issue, we propose a novel and comprehensive evaluation benchmark by employing a digital twin sampling approach. We construct three sets of datasets with complete "observations" that align with real-world historical human observation patterns.

### 3.2 DATA CONSTRUCTION

In this subsection, we provide a detailed description of the dataset construction procedures, including ❶ simulation data acquisition, ❷ real-world observation aggregation, ❸ spatiotemporal digital twin sampling, and ❹ spatial associations construction. Through the steps above, we integrate multiple databases that require specific domain knowledge to form an AI-ready public dataset.

**Simulation Data Acquisition**. In order to construct a virtual Earth with known dynamical behavior, we adopt three models from the widely recognized numerical simulation ensemble CMIP6[2] (Coupled Model Inter-comparison Project Phase 6), i.e., CESM2-omip1 (Danabasoglu et al., 2020), CESM2-omip2 (Danabasoglu et al., 2020) and GFDL-ESM4 (Dunne et al., 2020). For detailed information, please refer to Table 4 in the Appendix. These three models cover the entire Earth's historical records, with an annual temporal resolution and a spatial resolution of 1°×1° (latitude × longitude). The current resolution is sufficient for studying global-scale, long-term trends, which are central to understanding ocean ecosystem responses under climate change (Ito et al., 2025; Oschlies, 2021). Due to differences in simulation periods and iteration methods among the models, the reliable simulation periods for each model vary. In this study, the time span for CESM2-omip1 is from 1948 to 2009, for CESM2-omip2 it is from 1958 to 2018, and for GFDL-ESM4 it is from 1920 to 2014. This results in the acquisition of three virtual Earths, each with distinct operational dynamics and durations, parallel to the real Earth. The depth level is divided into 33 layers ranging from 0 to 5500 meters (Table 5).

---

[2]Access CMIP6 data from https://esgf-node.llnl.gov/search/cmip6/.

**Real-world Observation Aggregation**. To reflect the same data missing patterns on the real Earth, we aggregate in-situ observations of target variable (dissolved oxygen) and environmental variables from multiple public databases. The detailed sources of the observational data are listed in Table 3 of the Appendix. To ensure the validity of the data, we establish a unified quality control mapping rule. We map the quality FLAG settings of different databases to a common standard and retain only the "good quality" data for subsequent training and testing.

We follow the previous method (Schmidtko et al., 2017; He et al., 2019) to grid the observational data, where the temporal resolution, spatial resolution and the depth layer partitioning are consistent with the simulation data. Thus, we obtain a total of 1,999,268 ocean profile data, with the annual distribution of the number of profiles shown in Figure 4.

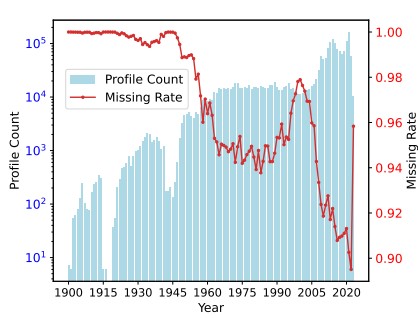

Figure 4: The number of profile data observed and missing rate each year in history (left y-axis is logarithmic scale).

**Spatiotemporal Digital Twin Sampling**. Furthermore, based on the in-situ observations, we obtain the corresponding four-dimensional spatiotemporal coordinates *(lon., lat., depth, time)*, and then project onto the virtual Earth for digital twin sampling. These synthetic "observations" are regarded as digital twins on another parallel Earth, which are usable for model training and enable the reconstruction of unobserved regions.

**Spatial Associations Construction**. Given the widespread application of graph neural networks (GNNs) in spatiotemporal data mining, we additionally construct dynamic spatial associations between data profiles as shown in Figure 5. We follow the previous work (Lu et al., 2024) to establish two types of neighbors, i.e., proximity neighbor and information hub, connecting both local and long-range spatial correlations (detailed in Appendix A.4). The graph modeling approach is more flexible. Proximity neighbors connect local spatial relationships by considering irregular geographic terrain, while information hubs connect unobserved nodes with distant observation nodes, capturing their relational dynamics.

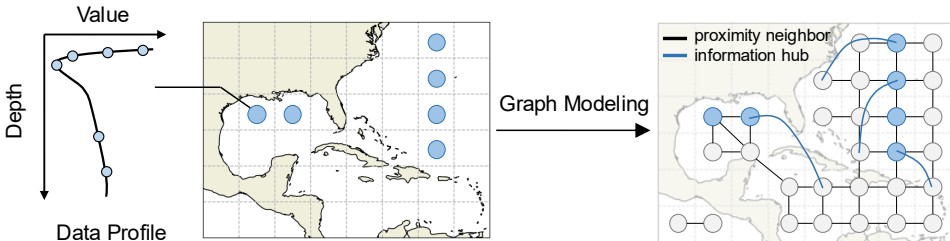

Figure 5: The construction of spatial associations within ocean data profiles.

## 4 EXPERIMENTS

To guide ML practitioners using OCEANVERSE datasets, this section provides several example machine learning workflows. The focus is to address the following research questions (**RQs**):

- **RQ1**: How do different machine learning methods perform on the three OCEANVERSE datasets?
- **RQ2**: Does the division of training, validation, and test data affect the model's performance?
- **RQ3**: How is the model's spatiotemporal generalization ability across different datasets?

### 4.1 EXPERIMENT SETUP

**Model Input and Output.** At time $t$, the model's input consists of the environmental variables at the current time $t$, as well as the target variables from time $t - T$ to $t + T$, which cover both historical and future time steps. It should be noted that since OCEANVERSE supports scientific data reconstruction

Table 2: Experiment results comparison of sparse 4D observation reconstruction on OCEANVERSE datasets. The best results are highlighted in bold, and the second best is underlined.

| Baseline | CESM2-omip1 | | CESM2-omip2 | | GFDL-ESM4 | |
|---|---|---|---|---|---|---|
| | RMSE ($\downarrow$) | R2 ($\uparrow$) | RMSE ($\downarrow$) | R2 ($\uparrow$) | RMSE ($\downarrow$) | R2 ($\uparrow$) |
| XGBoost | $0.074_{\pm 0.001}$ | $0.356_{\pm 0.018}$ | $0.076_{\pm 0.001}$ | $0.344_{\pm 0.025}$ | $0.072_{\pm 0.005}$ | $0.201_{\pm 0.104}$ |
| LSTM | $0.056_{\pm 0.005}$ | $0.628_{\pm 0.062}$ | $0.075_{\pm 0.017}$ | $0.319_{\pm 0.309}$ | $0.072_{\pm 0.004}$ | $0.225_{\pm 0.074}$ |
| MLP | $\underline{0.050}_{\pm 0.002}$ | $\underline{0.696}_{\pm 0.017}$ | $\underline{0.054}_{\pm 0.001}$ | $\underline{0.661}_{\pm 0.018}$ | $\underline{0.055}_{\pm 0.008}$ | $\underline{0.529}_{\pm 0.143}$ |
| Transformer | $0.069_{\pm 0.005}$ | $0.433_{\pm 0.082}$ | $0.074_{\pm 0.005}$ | $0.371_{\pm 0.076}$ | $0.072_{\pm 0.001}$ | $0.216_{\pm 0.020}$ |
| GRIN | OOM | OOM | OOM | OOM | OOM | OOM |
| ImputeFormer | $0.087_{\pm 0.000}$ | $0.101_{\pm 0.002}$ | $0.089_{\pm 0.000}$ | $0.094_{\pm 0.004}$ | $0.081_{\pm 0.000}$ | $0.020_{\pm 0.002}$ |
| TIDER | $0.087_{\pm 0.000}$ | $0.105_{\pm 0.001}$ | $0.088_{\pm 0.000}$ | $0.111_{\pm 0.002}$ | $0.072_{\pm 0.000}$ | $0.223_{\pm 0.005}$ |
| OxyGenerator | $\mathbf{0.044}_{\pm 0.005}$ | $\mathbf{0.764}_{\pm 0.052}$ | $\mathbf{0.049}_{\pm 0.005}$ | $\mathbf{0.716}_{\pm 0.061}$ | $\mathbf{0.053}_{\pm 0.003}$ | $\mathbf{0.579}_{\pm 0.043}$ |

task, future data beyond time $t$ can also be utilized to capture the temporal patterns of the data, thus aiding in the reconstruction of the target variable. The model's output is the target variable at time $t$.

**Baselines**. Eight baselines are used to evaluate the reconstruction performance on OCEANVERSE. These include 4 classic machine learning methods: XGBoost, MLP, LSTM, and Transformer. These models perform regression based on current timestep environmental variables and target variable time series. LSTM and Transformer models further enhancing the learning of temporal features. Meanwhile, four recent models—GRIN (Cini et al., 2022), ImputeFormer (Nie et al., 2024), TIDER (Liu et al., 2023b), and OxyGenerator (Lu et al., 2024)—are also included as baselines for comparison. Notably, point-based time series imputation models, like CSDI (Tashiro et al., 2021) and BRITS (Cao et al., 2018), are not included for comparison. The historical oceanic profile data exhibits high sparsity, with a large number of samples with no historical observations at all, making it impossible to use as baselines. For detailed information, please refer to Appendix B.2.

**Dataset Split**. Dataset Split: We split the observation data into training and validation sets based on time. The first 70% of the years are used for training, while the remaining 30% are used for validation. Specifically, for the CESM2-omip1 dataset, the training period is from 1948 to 1991 (the first 44 years), and the validation period is from 1992 to 2009. For CESM2-omip2, the training period is from 1958 to 2000 (the first 43 years), and the validation period is from 2001 to 2018. For GFDL-ESM4, the training period is from 1920 to 1986 (the first 67 years), and the validation period is from 1987 to 2014. During the model testing phase, the evaluation focuses on the reconstruction results for all unobserved regions, reflecting the model's spatio-temporal global reconstruction performance.

**Evaluation Metrics**. Our evaluation metrics are computed separately for the target variable (i.e., dissolved oxygen) in the output vector. The Root Mean Square Error (RMSE) and the coefficient of determination ($R^2$) are calculated independently for each horizontal and vertical location, and subsequently averaged both horizontally and vertically to produce the summary statistics. Additionally, other regression evaluation metrics are readily computed for further assessment. All experiments are repeated five times, and the reported results include the mean and variance.

## 4.2 EXPERIMENT RESULTS

**Overall performance (RQ1)**. Table 2 presents the performance of different baselines across the three OCEANVERSE datasets, and we obtain the following observations: ❶ OxyGenerator and MLP consistently rank first and second, respectively. OxyGenerator demonstrates superior performance due to its well-designed spatio-temporal graph neural networks, which embed both the target variable time series and environmental variables. MLP achieves suboptimal results through the training of a large number of parameters, surpassing several more complex models. An important reason for this lies in the mismatch between model complexity and the amount of data available. Due to the high data sparsity, many sophisticated models struggle to learn the temporal and spatial correlations within the data, thereby limiting their expressive power and showing inferior performance. ❷ XGBoost, LSTM, and Transformer show significant performance fluctuations across different datasets. XGBoost, as a tree-based boosting model, benefits from the flexibility of decision tree splitting rules in handling missing values, which makes it better to many other models. Although LSTM and Transformer are well-suited for capturing temporal features, they struggle to effectively handle time series with a

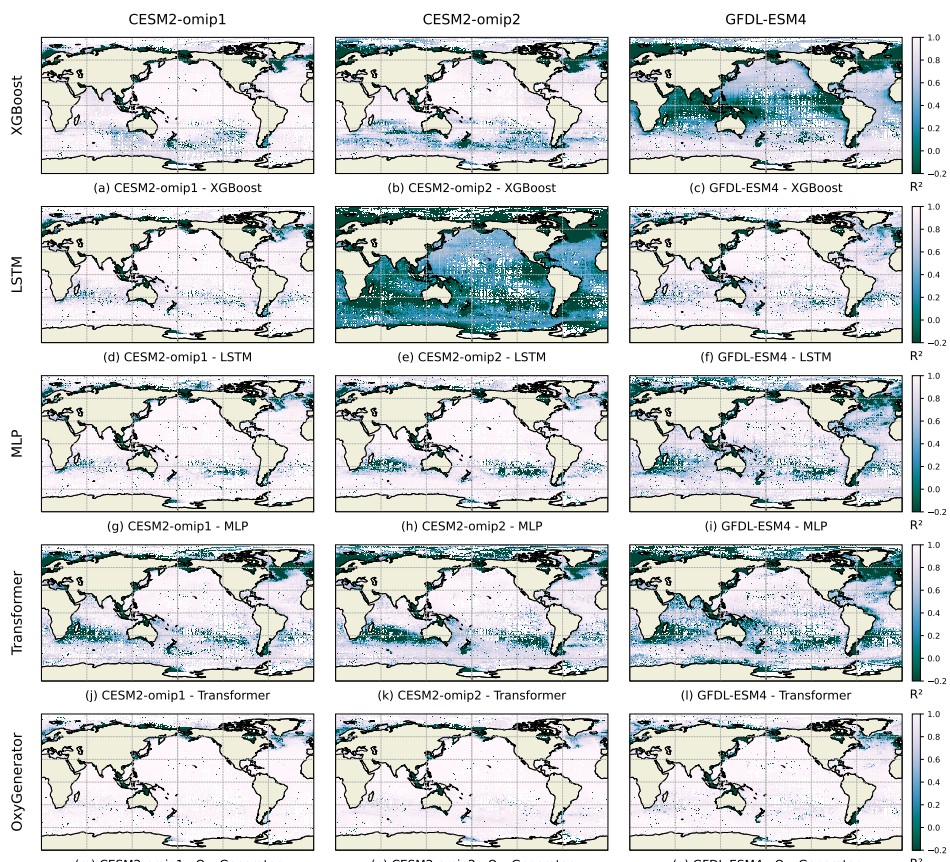

Figure 6: Spatial variation of model performance across three OCEANVERSE datasets, with darker colors indicating poorer reconstruction performance.

large amount of missing data, leading to considerable performance variability. ❸ ImputeFormer and TIDER perform poorly under conditions of large-scale missing data. Additionally, GRIN faces an out-of-memory (OOM) issue when performing operations on large-scale graphs, as it requires computing the entire adjacency matrix.

**Model performance over global ocean (RQ1)**. Figure 6 illustrates the spatial distribution of reconstruction errors for different models across global oceanic regions. It can be shown that the models generally exhibit poorer reconstruction performance in coastal regions compared to open ocean areas. Nearshore regions are influenced by more complex human activities and exhibit more intense variation patterns. Additionally, the uncertainty is greater in the Southern Hemisphere, as there are fewer observational data available compared to the Northern Hemisphere, which leads to less effective learning by the neural networks.

**Model performance over time (RQ1)**. Figure 7 illustrates the performance variation of five representative models across the time dimension. Following a significant increase in human observations after the 21st century, the availability of more observational data positively impacts the reconstruction performance, with a particularly notable performance improvement of Transformer.

**The impact of different data partitions on model training (RQ2)**. The spatiotemporal nature of the data means that partitioning the training and validation datasets can influence the selection of optimal model parameters. We compare three partitioning methods: ❶ *Random Split*: The datasets are randomly divided in a 70:30 ratio. ❷ *Temporal Split*: The first 70% of the years are used for training, and the remaining 30% for validation. ❸ *Spatial Split*: Based on the World Ocean Database, the global oceans are divided into five regions: Atlantic, Pacific, Indian Oceans, Polar Regions, and Enclosed Seas, with a 70:30 split. Figure 8 shows the performance of dif-

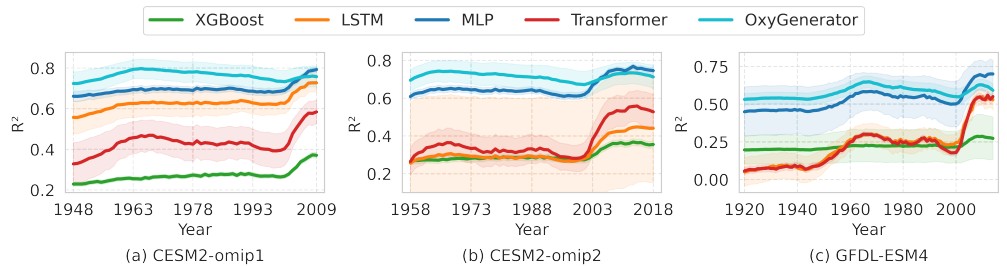

Figure 7: Temporal variation of model performance across three OCEANVERSE datasets.

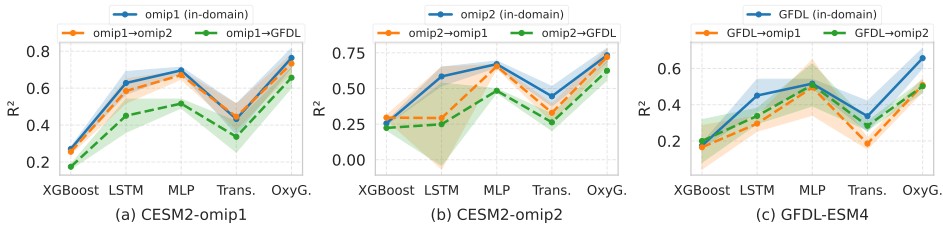

Figure 9: Generalization performance of different models across virtual Earth.

ferent models under various data split schemes. Overall, temporal split yields the best results, followed by spatial split, with random split showing the worst performance. Random split introduces local spatiotemporal autocorrelation in both the training and validation datasets, which interferes with model selection. Space split provides generalizable validation across distinct marine regions, but the model struggles to learn features from regions it has not encountered during training. Therefore, we recommend that ML practitioners use temporal split for data partitioning.

**Model generalization ability across different virtual Earth (RQ3).** The ability of a model trained on one virtual Earth to be directly applied to other virtual Earth depends on its generalization capacity. Our OCEANVERSE provides three different simulation conditions, and Figure 9 illustrates the model's performance in both in-domain and out-of-domain settings. omip1 and omip2 are two generations of numerical simulations developed by CESM2, leading to better mutual generalization compared to GFDL-ESM4. Additionally, we observe that the generalization performance of models such as LSTM and MLP shows more variability.

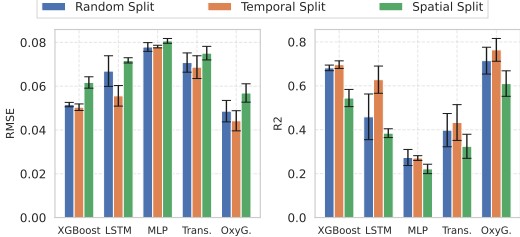

Figure 8: Comparison of model performance under different data split scenarios.

## 5 LIMITATIONS

**Discrepancy between real-world data and numerical simulation.** While real-world observations provide the most practical first-hand data, the missing value in historical data can never be measured, and thus, we cannot obtain a definitive answer for evaluation. To ensure that our benchmark can be applied to the real Earth, we provide three sets of virtual Earths through simulations, which are widely recognized by ocean scientists and have been adjusted using real-world observations. We hope that future researchers will improve the model's representation power and generalization ability with our OCEANVERSE. OCEANVERSE will continue to track the latest simulation models and facilitate more informed model selection through the provision of multiple sets for joint comparison.

**Higher Spatiotemporal Resolution for Scientific Reconstruction.** OCEANVERSE currently provides data with a spatial resolution of $1° \times 1°$ and a temporal resolution of 1 year. This level of

spatiotemporal resolution is sufficient and appropriate for analyzing long-term global-scale trends. Expanding to finer spatiotemporal resolutions would better support the analysis of small-scale dynamic processes. However, such an approach would also exacerbate data sparsity and heterogeneity, thereby increasing the challenges for AI model algorithms.

## 6 CONCLUSION

In this paper, we present an AI-ready evaluable ocean science dataset OCEANVERSE, aimed at contributing advanced machine learning methods for reconstructing sparse 4D ocean observations, thereby enabling a better understanding of the impacts of climate change and human activities on the ocean ecosystem. To facilitate efficient use, both the OCEANVERSE dataset and the baseline code are open-source. In the future, we plan to further expand the components of OCEANVERSE, extending beyond core oceanic elements such as dissolved oxygen to include additional elements like dissolved inorganic phosphorus, lead, and mercury, thereby promoting the development of AI for Ocean.

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

## A  ADDITIONAL DETAILS OF OCEANVERSE CONSTRUCTION

### A.1  OBSERVATION DATA

Ocean profile data relies on field measurements from cruises, buoys, CTD sensors, and other sources, resulting in a dispersed nature of the data. Table 3 provides detailed information about the multi-source observation databases, from which we have gathered a total of 1,999,268 ocean profile data points from five different databases. Figure 10 shows the gridded observation data. The color bar displays the proportion(%) of available observed data for each grid. The gridded data shows significant sparse and uneven distribution. There is a noticeable increase in data volume after 1955-1959.

- **World Ocean Database (WOD)**: WOD is world's largest collection of uniformly formatted, quality controlled, publicly available ocean profile data. It is a powerful tool for oceanographic, climatic, and environmental research, and the end result of more than 20 years of coordinated efforts to incorporate data from institutions, agencies, individual researchers, and data recovery initiatives into a single database.

- **CLIVAR and Carbon Hydrographic Data Office (CCHDO)**: CCHDO supports oceanographic research by providing access to high quality, global, vessel-based CTD and hydrographic data from GO-SHIP, WOCE, CLIVAR and other repeat hydrography programs. These data are openly accessible and served in standardized community formats (WHP-Exchange, WOCE, and netCDF).

- **Argo**: Argo is an international program that collects information from inside the ocean using a fleet of robotic instruments that drift with the ocean currents and move up and down between the surface and a mid-water level. Each instrument (float) spends almost all its life below the surface.

- **Global Ocean Data Analysis Project version2.2022 (GLODAP)**: GLODAP is a synthesis activity for ocean surface to bottom biogeochemical data collected through chemical analysis of water samples. GLODAP is publicly available, discoverable, and citable. GLODAP enables quantification of the ocean carbon sink, ocean acidification and evaluation of ocean biogeochemical models.

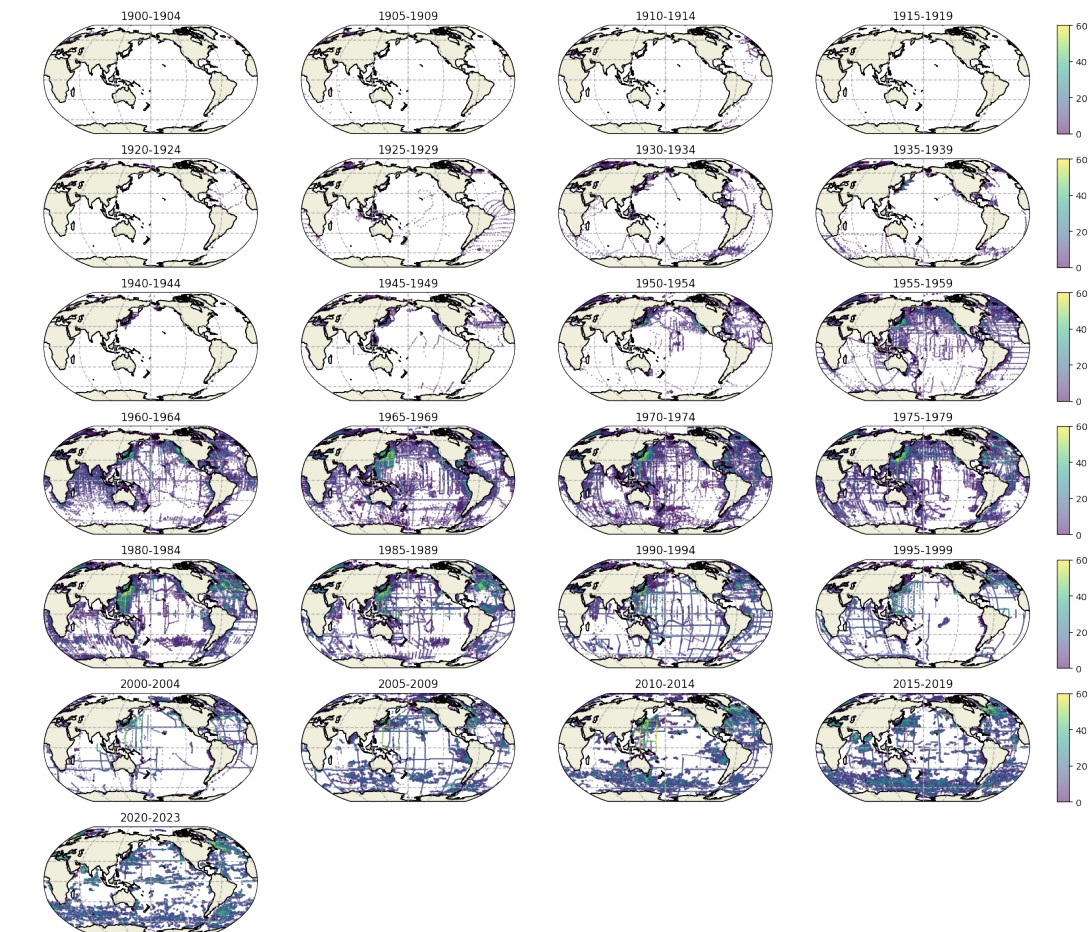

Figure 10: Spatial distribution of dissolved oxygen gridded observation data every five years.

- **Geotraces IDP**: GEOTRACES intermediate data product (IDP) gathers hydrographic and marine biogeochemical data acquired on 89 cruises. The IDP contains trace elements that serve as micronutrients, tracers of continental sources to the ocean (e.g., aerosols and boundary exchange), contaminants, radioactive and stable isotopes used in paleoceanography and a broad suite of hydrographic parameters used to trace water masses, as well as, it provides biological data.

Table 3: Detailed Information of data sources for global ocean observations.

| Database | Time | Institution | Source | Access Date |
|---|---|---|---|---|
| World Ocean Database (WOD 2018) | 1900-2023 | National Centers for Environmental Information | https://www.ncei.noaa.gov/ | 2023-05 |
| CLIVAR and Carbon Hydrographic Data Office (CCHDO) | 1922-2023 | CLIVAR and Carbon Hydrographic Data Office | https://cchdo.ucsd.edu/ | 2023-05 |
| Argo | 2001-2023 | Argo Global Data Assembly Center | https://argo.ucsd.edu/ | 2023-05 |
| Global Ocean Data Analysis Project version2.2022 (GLODAPV2_2022) | 1972-2021 | NOAA's National Centers for Environmental Information (NCEI) | https://glodap.info/ | 2023-05 |
| Geotraces IDP | 2007-2018 | GEOTRACES International Data Assembly Centre (GDAC) | https://www.geotraces.org | 2023-10 |

Table 4: The statistical information and comparison of different numerical simulation methods

| Model Name | CESM2-OMIP1 | CESM2-OMIP2 | GFDL-ESM4 |
|---|---|---|---|
| Developing Institution | National Center for Atmospheric Research (NCAR) | National Center for Atmospheric Research (NCAR) | Geophysical Fluid Dynamics Laboratory (GFDL) |
| Spatial Range | Global scale | Global scale | Global scale |
| Spatial Resolution | $1° \times 1°$ | $1° \times 1°$ | $1° \times 1°$ |
| Temporal Range | 1948-2009 | 1958-2018 | 1920-2014 |
| Temporal Resolution | Yearly output | Yearly output | Yearly output |
| Simulation Assumptions | Fixed greenhouse gas concentrations or specific scenarios (e.g., RCP8.5) | Fixed greenhouse gas concentrations or specific scenarios (e.g., RCP8.5) | Fixed greenhouse gas concentrations or specific scenarios (e.g., SSP5-8.5) |
| Simulation Conditions | Global climate change scenarios, focusing on ocean carbon cycle and ecosystems | Global climate change scenarios, focusing on ocean carbon cycle and ecosystems | Global climate change scenarios, focusing on carbon cycle, ocean acidification, and ecosystem processes |
| Ocean Biogeochemical Processes | Models carbon, nitrogen, phosphorus cycles, focusing on biogeochemical feedbacks | More detailed biogeochemical process simulations, focusing on ocean carbon cycle and nutrient exchange | Detailed simulation of ocean carbon cycle, ocean acidification, and ecosystem responses |
| Key Research Areas | Climate change impacts on marine ecosystems, carbon absorption, ocean productivity | Climate change impacts on marine ecosystems, biogeochemical feedbacks | Climate change, ocean carbon absorption, ocean acidification, and ecosystem impacts |
| Output Data | Yearly data on ocean carbon cycle, ocean productivity, biogeochemical data | Yearly data on ocean carbon cycle, ocean productivity, biogeochemical data | Yearly data on ocean carbon cycle, greenhouse gas exchanges, ocean acidification |
| Primary Use | Supports ocean biogeochemical research, ocean carbon sink assessment | Supports ocean biogeochemical research, ecosystem response to climate change | Supports climate change, ocean acidification, and carbon cycle research, global warming impact assessments |

## A.2 NUMERICAL SIMULATION

The CMIP6 (Coupled Model Intercomparison Project Phase 6) is the sixth phase of a global collaboration among climate modeling groups to improve understanding of climate change and its impacts. It provides a framework for comparing and evaluating climate models, particularly those used to simulate future climate scenarios.

In CMIP6, models simulate a wide range of climate variables across different emissions scenarios, including temperature, precipitation, ice sheets, and ocean conditions. One of the key aspects of CMIP6 is the inclusion of biogeochemical processes, such as the simulation of dissolved oxygen (DO) in the oceans. Models in CMIP6 aim to simulate how ocean oxygen levels change in response to climate change, including factors such as warming, acidification, and changes in ocean circulation. In Table 4, we present the dissolved oxygen simulation model used in CMIP6 in this paper.

- **CESM2 omip1**: The Community Earth System Model (CESM) is a climate/Earth system coupled model used to simulate past, present, and future climates. The ocean component of CESM2 undergoes various physical model and numerical computation improvements, while utilizing the Marine Biogeochemistry Library (MARBL) to represent ocean biogeochemistry. The experiment, omip1, is driven by the CORE-II (Coordinated Ocean - ice Reference Experiments) atmospheric data, and initialized with physical and biogeochemical ocean observations to conduct ocean dissolved oxygen simulations.

- **CESM2 omip2**: Experiment omip2 shares the same model configuration as omip1 but is forced by the JRA-55 atmospheric data with higher spatial resolution than CORE-II.

- **GFDL-ESM4**: The Earth System Model4 (ESM4) is the fourth-generation chemistry-carbon-climate coupled climate model developed by the Geophysical Fluid Dynamics Laboratory (GFDL). Compared to previous versions, this model more comprehensively represents chemical cycling and ecosystems. Particularly, the model incorporates interactions between ocean ecology and biogeochemistry. The historical experiment utilizes rich climate observational data from 1850 to the present, imposing environmental change conditions consistent with observations to obtain historical simulations of ocean dissolved oxygen.

Table 5: Dividing the ocean from the surface to 5,500 meters into 33 depth layers.

| Layer ID | Depth Level (m) | Level Boundary (m) | |
|---|---|---|---|
| 1 | 0 | 0 | 5 |
| 2 | 10 | 5 | 15 |
| 3 | 20 | 15 | 25 |
| 4 | 30 | 25 | 40 |
| 5 | 50 | 40 | 62.5 |
| 6 | 75 | 62.5 | 87.5 |
| 7 | 100 | 87.5 | 112.5 |
| 8 | 125 | 112.5 | 137.5 |
| 9 | 150 | 137.5 | 175 |
| 10 | 200 | 175 | 225 |
| 11 | 250 | 225 | 275 |
| 12 | 300 | 275 | 350 |
| 13 | 400 | 350 | 450 |
| 14 | 500 | 450 | 550 |
| 15 | 600 | 550 | 650 |
| 16 | 700 | 650 | 750 |
| 17 | 800 | 750 | 850 |
| 18 | 900 | 850 | 950 |
| 19 | 1000 | 950 | 1050 |
| 20 | 1100 | 1050 | 1150 |
| 21 | 1200 | 1150 | 1250 |
| 22 | 1300 | 1250 | 1350 |
| 23 | 1400 | 1350 | 1450 |
| 24 | 1500 | 1450 | 1625 |
| 25 | 1750 | 1625 | 1875 |
| 26 | 2000 | 1875 | 2250 |
| 27 | 2500 | 2250 | 2750 |
| 28 | 3000 | 2750 | 3250 |
| 29 | 3500 | 3250 | 3750 |
| 30 | 4000 | 3750 | 4250 |
| 31 | 4500 | 4250 | 4750 |
| 32 | 5000 | 4750 | 5250 |
| 33 | 5500 | 5250 | 5500 |

## A.3 DATA GRIDDING

We follow the method described in previous studies (Schmidtko et al., 2017; He et al., 2019) to grid the observational data, ensuring a consistent and structured representation of the data across both time and space. The temporal resolution is set to annual, allowing us to capture long-term trends and patterns in the data. For spatial resolution, we use a grid with a resolution of $1° \times 1°$ (latitude × longitude), providing a reasonable level of detail while ensuring computational feasibility. The depth profile is divided into 33 distinct layers, ranging from the surface down to 5500 meters, with each layer representing a specific depth interval, as detailed in Table 5. To mitigate any potential biases arising from data clustering, we apply a median-binning approach to all profiles. Specifically, data points within a $0.25°$ window in both latitude and longitude, and within a 3-month temporal window, are grouped together and the median value is used for each bin. This approach helps smooth out variations due to localized data clusters, providing a more robust representation of the underlying spatial and temporal patterns.

### A.4 GRAPH MODELING

We represent dissolved oxygen profiles as nodes $\mathbf{V}$ in the graph $\mathbf{G}$, and the relationships between the dissolved oxygen profiles as edges $\mathbf{E}$. We further consider two types of edge relationships, defined as proximity neighbors and information hubs:

- **Proximity Neighbors.** Proximity neighbors are based on the *First Law of Geography*, which states that "everything is related to everything else, but near things are more related than distant things." This law suggests that nearby locations exhibit similar characteristics. Thus, we consider the proximity neighbors as the data grids within a small range, e.g., $1°$ in both latitude and longitude. Regarding the irregular boundaries, we adopt the bedrock elevation data to ensure the rationality.

- **Information Hubs.** Information hubs refer to nodes that, although geographically distant, possess rich observational information. We define the observation completeness $C_{\text{obs}}$ over a time span of $T$ timesteps before and after the current moment, i.e., $C_{\text{obs}} = \frac{\|\omega_{i,j,d,t-T:t+T}\|_1}{2T} = \frac{\sum_\tau \|\omega_{i,j,d,\tau}\|}{2T}$. When the observation completeness $C_{\text{obs}}$ exceeds the threshold $c_0$, nodes within a distance of up to $5°$ are regarded as information hubs.

## B ADDITIONAL DETAILS OF EXPERIMENTS

### B.1 DATA SPLIT

We partition the observational data into training and validation sets based on three distinct methodologies: random, time, and space.

- **Random split**: The training and validation sets are randomly divided in a 7:3 ratio, while the test data spans all years. We employ fixed random seeds to ensure the reproducibility of the dataset partitioning process.

- **Temporal split**: The observation data is divided into training and validation sets based on time. The first 70% of the years are used for training, while the remaining 30% are used for validation. Specifically, for the CESM2-omip1 dataset, the training period is from 1948 to 1991 (the first 44 years), and the validation period is from 1992 to 2009. For CESM2-omip2, the training period is from 1958 to 2000 (the first 43 years), and the validation period is from 2001 to 2018. For GFDL-ESM4, the training period is from 1920 to 1986 (the first 67 years), and the validation period is from 1987 to 2014. During the model testing phase, the evaluation focuses on the reconstruction results for all unobserved regions, reflecting the model's spatio-temporal global reconstruction performance.

- **Spatial split**: Based on the range mask from WOD (World Ocean Database), the global ocean is divided into five regions: the Atlantic Ocean, the Pacific Ocean, the Indian Ocean, the Polar Oceans, and the enclosed seas. The training and validation sets are then allocated in a 7:3 ratio according to these spatial regions. The detailed division is shown in Table 6.

### B.2 BASELINES

In this paper, we use eight baselines for task evaluation on the OCEANVERSE dataset. In addition to the four commonly used classical models, XGBoost, MLP, LSTM, and Transformer, we also introduce four recent research advancements:

- **GRIN** (Cini et al., 2022): A message-passing-based bidirectional recurrent neural network designed for spatio-temporal imputation.

- **ImputeFormer** (Nie et al., 2024): A low-rank-induced Transformer model that achieves a balance between signal and noise for general spatio-temporal imputation.

- **TIDER** (Liu et al., 2023b): A matrix factorization-based method that employs disentangled neural representations to model complex dynamics in multivariate time series.

- **OxyGenerator** (Lu et al., 2024): A graph-based method for reconstructing oxygen levels, incorporating zoning-varying spatial correlations and chemistry-informed regularization techniques.

Table 6: Ocean Regions and Their Corresponding Train and Validation Sets

| Ocean Regions | Train | Validation |
|---|---|---|
| **Atlantic Ocean** | 1. North Atlantic | 1. Equatorial Atlantic |
| | 2. Coastal N Atlantic | |
| | 3. South Atlantic | |
| | 4. Coastal S Atlantic | |
| **Pacific Ocean** | 5. North Pacific | 3. Equatorial Pacific |
| | 6. Coastal N Pacific | 4. Coastal Eq Pacific |
| | 7. South Pacific | |
| | 8. Coastal S Pacific | |
| **Indian Ocean** | 9. North Indian | 5. Equatorial Indian |
| | 10. Coastal N Indian | 6. Coastal Eq Indian |
| | 11. South Indian | |
| | 12. Coastal S Indian | |
| **Polar Oceans** | 13. Arctic | 7. Antarctic |
| **Enclosed Seas** | 14. Baltic Sea | 8. Mediterranean |
| | 15. Red Sea | 9. Black Sea |
| | | 10. Persian Gulf |
| | | 11. Sulu Sea |

## B.3 COMPUTE RESOURCES

All the evaluated models are implemented on a server with 128 CPUs (AMD EPYC 7542) and 8 GPUs (NVIDIA GTX 4090, 24GB memory).

## B.4 EVALUTION METRICS

In this study, we evaluate the performance of the sparse observation reconstruction using two commonly employed metrics: Root Mean Square Error (RMSE) and the coefficient of determination ($R^2$). The RMSE is defined as the square root of the average squared differences between predicted and observed values, providing a measure of the accuracy of the model's predictions. Mathematically, it is expressed as:

$$RMSE = \sqrt{\frac{1}{n}\sum_{i=1}^{n}(y_i - \hat{y}_i)^2}$$

where $y_i$ represents the true values and $\hat{y}_i$ are the predicted values. A lower RMSE indicates better predictive accuracy. On the other hand, $R^2$ quantifies the proportion of variance in the dependent variable that is explained by the model. It is calculated as:

$$R^2 = 1 - \frac{\sum_{i=1}^{n}(y_i - \hat{y}_i)^2}{\sum_{i=1}^{n}(y_i - \bar{y})^2}$$

where $\bar{y}$ is the mean of the observed values. The value of $R^2$ typically ranges from 0 to 1, with a value closer to 1 indicating a better fit of the model to the data. However, $R^2$ can sometimes be less than 0, which occurs when the model performs worse than simply predicting the mean of the target values for all observations. In such cases, the model fails to capture the variance in the data, indicating particularly poor performance.

## B.5    CASE STUDY FOR RECONSTRUCTION RESULTS

Due to time constraints, we evaluate OxyGenerator by performing reconstructions on real-world observational data. We show the results of six depth layers from the year 1980, specifically at depths of 0 m, 100 m, 500 m, 1000 m, 2000 m and 5000 m.

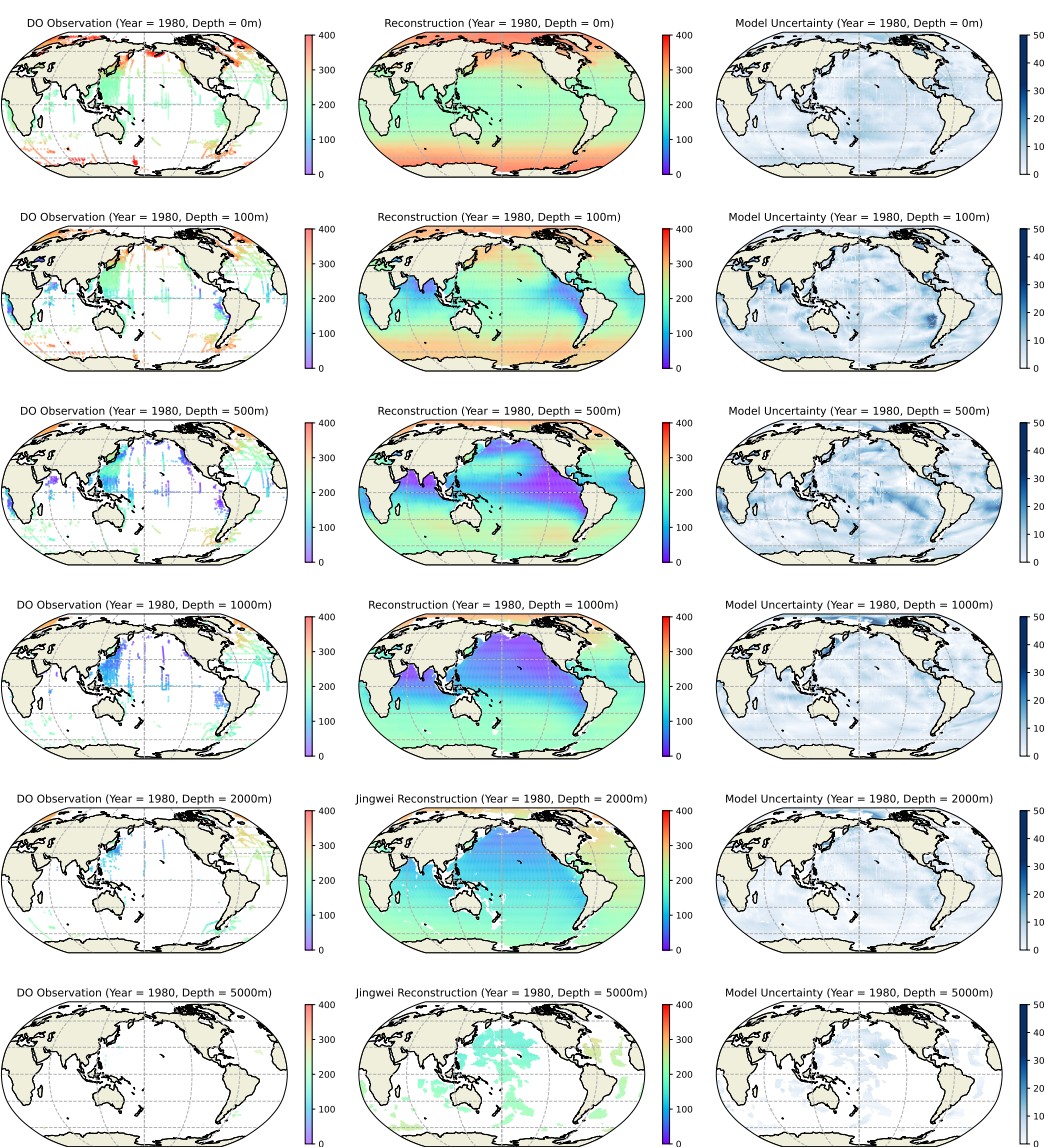

Figure 11: Cases of dissolved oxygen observations, reconstruction results, and model uncertainty.

## B.6    DIFFERENCES BETWEEN OBSERVATIONS AND MODEL-BASED DATA

To better understand the discrepancies between observational datasets and Earth system model simulations, we examine their differences from both spatial and temporal perspectives. The comparison highlights systematic biases as well as model-dependent behaviours across depth layers.

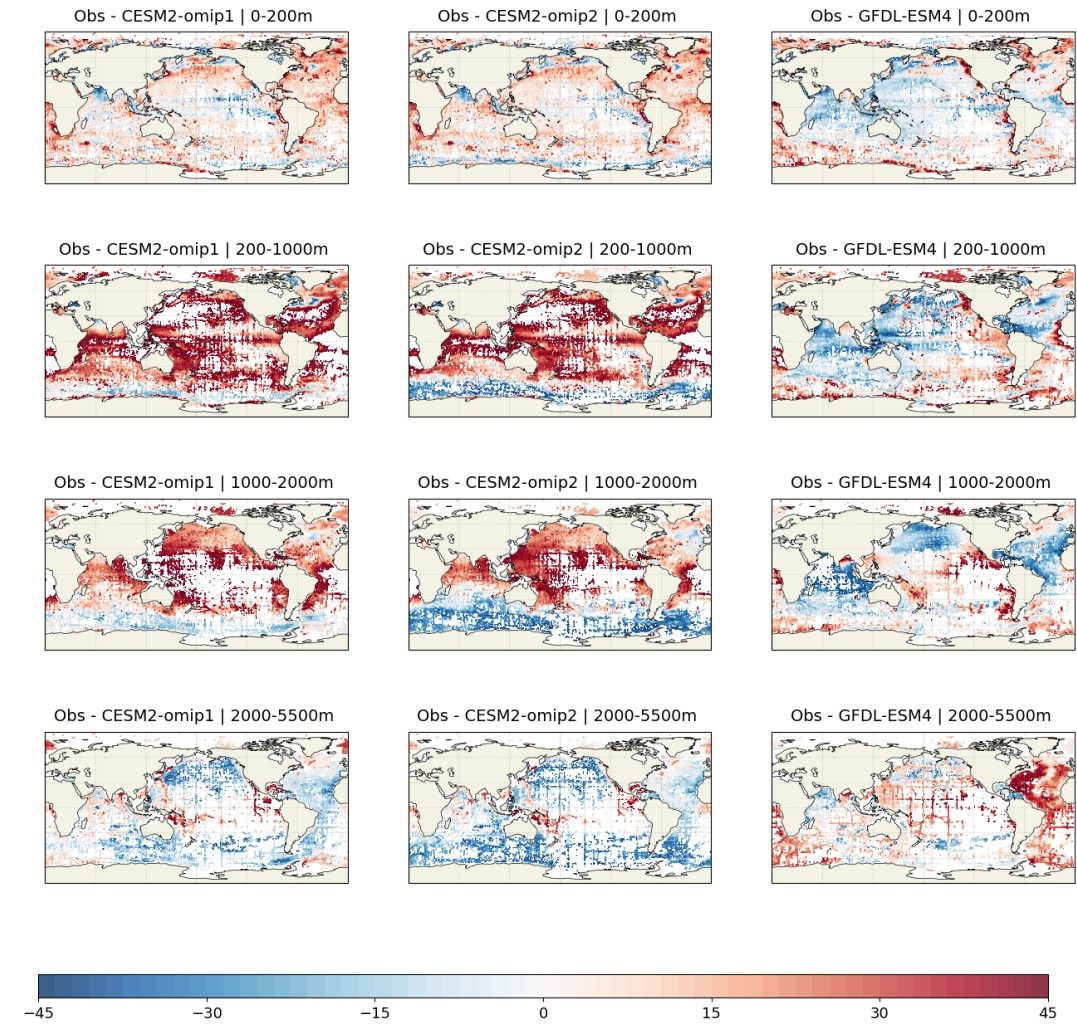

Figure 12: Spatial Differences between observation and numerical simulations.

**Spatial Difference**. Figure 12 presents the differences between observations and three model simulations across four depth layers (0–200 m, 200–1000 m, 1000–2000 m, and 2000–5500 m), where red indicates regions with observational oxygen higher than model values, and blue indicates the opposite. Clear model-to-model discrepancies are evident, as each model exhibits distinct geographical patterns and magnitudes of bias relative to observations. Overall, the upper-ocean layer (0–200 m) shows more scattered and heterogeneous biases, whereas the mid-depth layers (200–2000 m) display more coherent, banded structures. In the deep ocean (2000–5500 m), the bias field becomes relatively uniform, though the sign of the bias (positive vs. negative) differs substantially among models, reflecting variations in their internal physical and biogeochemical representations.

**Temporal Difference**. Figure 13 shows the temporal distribution of differences between observation and numerical simulation oxygen across depth layers. The upper ocean (0–200 m) exhibits larger fluctuations and more dispersed anomalies, whereas differences become progressively more stable with increasing depth. For CESM2-OMIP1 and CESM2-OMIP2, observations generally show higher oxygen concentrations than the models in the 0–2000 m layers, while the models produce higher oxygen than observed in the 2000–5500 m deep ocean. In contrast, GFDL-ESM4 displays the opposite pattern, with model values exceeding observations in the upper and mid-depth layers but falling below observations in the deep ocean.

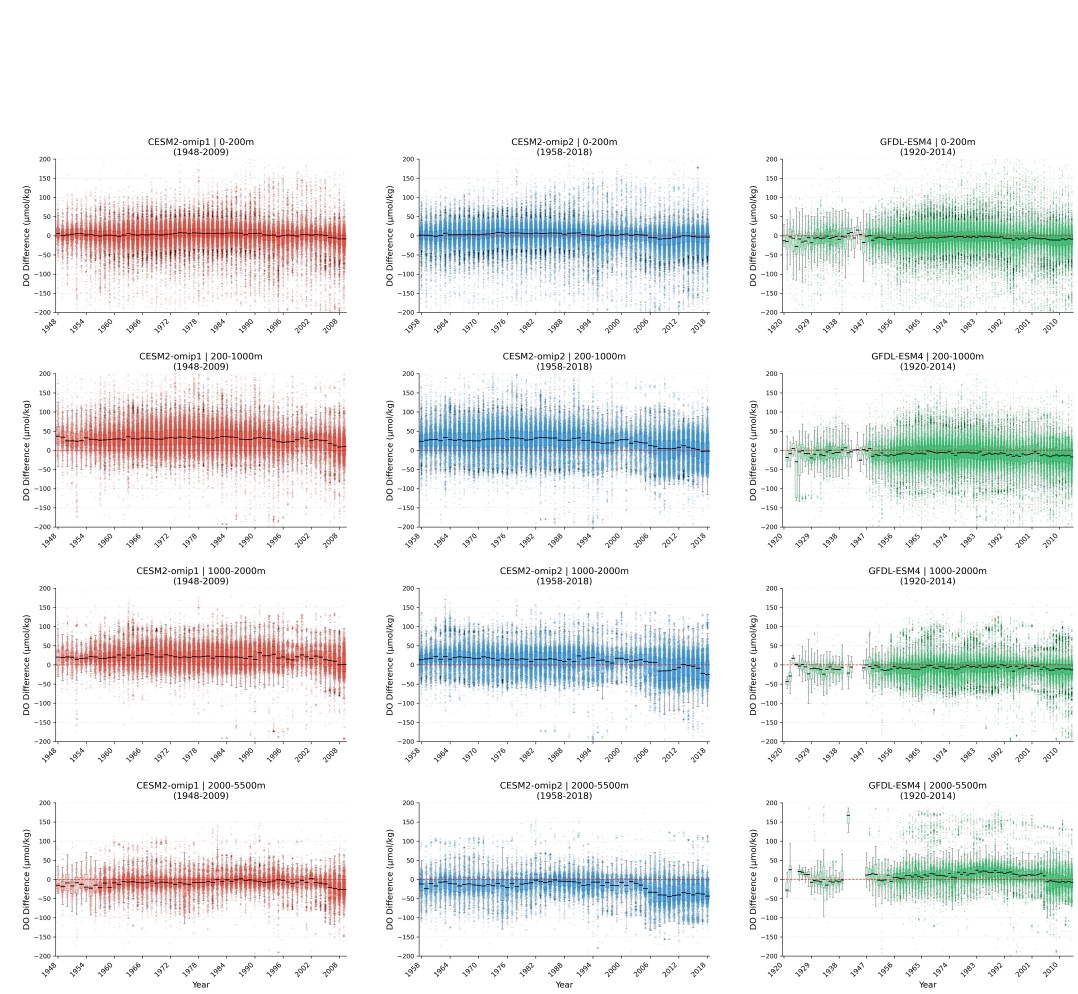

Figure 13: Temporal Differences between observation and numerical simulations.