# OpenReview forum: "OceanVerse: Evaluable 4D Ocean Element Reconstruction Dataset under Realistic Sparsity"
_ICLR.cc/2026/Conference — Submitted to ICLR 2026_

### Official Review · Reviewer_QQvM · 2025-10-26

**Soundness:** 3
**Presentation:** 3
**Contribution:** 2
**Rating:** 4
**Confidence:** 4

**Summary:**

This paper introduces OCEANVERSE, an evaluable 4D ocean reconstruction benchmark for studying learning-based data assimilation under realistic observational sparsity. Using three numerical Earth simulations (CESM2-OMIP1/2, GFDL-ESM4) as background truth, the authors construct a spatiotemporal dataset where real-world observation distributions are used to generate MNAR sampling masks. They define a unified reconstruction task, establish an evaluation framework, and benchmark several models including classical regressors, sequence models, and a spatio-temporal GNN.

**Strengths:**

(1) Important questions to AI for Oceanography. The paper addresses an under-explored yet crucial area — AI-driven ocean field reconstruction and data assimilation. This is an important and long-standing challenge in marine science, as ocean observations are extremely sparse and irregular in both space and time. By constructing a complex and large-scale benchmark under realistic observational sparsity, the work provides a much-needed foundation for developing and comparing machine learning models in this domain. This contribution could meaningfully advance the emerging field of AI for Oceanography, fostering the development of more effective assimilation and reconstruction systems.

(2) Realistic sampling design. The proposed Digital Twin Sampling method is well-motivated and technically elegant. It leverages real in-situ observation distributions (from ∼2M historical profiles) to generate non-random, observation-like sampling masks (MNAR), while maintaining full evaluation capability through simulated background fields. This design realistically mimics the spatiotemporal coverage of ocean observations and enables quantitative benchmarking — a key advantage over prior synthetic or randomly masked datasets.

**Weaknesses:**

(1) Although the dataset is large and technically well-structured, its overall design remains much simpler than real-world data assimilation systems. The spatial and temporal resolutions are relatively coarse, and only a single variable (dissolved oxygen) is included. As a result, the benchmark captures statistical reconstruction behavior but does not reflect the full complexity of realistic ocean assimilation processes.

(2) The assessment is performed entirely within digital-twin simulations, comparing reconstructions against GCM-generated fields rather than real in-situ or reanalysis observations. This makes the evaluation quantitatively consistent but scientifically detached — models may align closely with the numerical model’s climatology while providing limited evidence of real-world assimilation skill or forecasting utility.

**Questions:**

(1) Could the authors elaborate on how the OCEANVERSE setup relates to real ocean data assimilation systems? In particular, to what extent can the current benchmark (based on digital-twin simulations) reflect or predict model performance in real ocean reconstruction or assimilation tasks? What assumptions or simplifications might limit this correspondence?

(2) What was the rationale for selecting dissolved oxygen as the sole reconstruction target? Does it have specific scientific significance? Would the framework generalize well to other physical variables such as temperature or salinity?

---

> ### Author Response · Authors · 2025-11-23
>
> We sincerely appreciate your thoughtful review and your recognition that *this paper addresses an "under-explored yet crucial" problem*. The challenge of reconstructing ocean variables from sparse observations is indeed an *"important and long-standing issue in marine science"*, and the field has lacked an effective evaluation benchmark for many years. We are grateful that you found our proposed solution to be *"well-motivated and technically elegant"*.
> We hope that our responses below will address your concerns clearly, and we would be very grateful to receive your support.
>
> **(W1, Q1)** Thank you for raising this point. We would like to clarify that OceanVerse is not intended to replicate a full data assimilation system, nor do we aim to perform data assimilation. Instead, our goal is to provide a benchmark for AI-based reconstruction under real-world sparse ocean observations, which is a distinct but complementary objective.
>
> The current benchmark is not a purely statistical reconstruction task. Rather, it reflects global-scale, long-term 4D dissolved-oxygen variability, which is a central topic in climate change research and an area where substantial scientific debate persists precisely due to observational scarcity. For these scientific questions, the 1° × 1° and annual resolution is already sufficient and commonly used in major global biogeochemical studies.
>
> We fully agree that more variables and finer resolutions would bring the benchmark closer to the complexity of operational data assimilation systems. Indeed, extending OceanVerse to additional biogeochemical variables and higher-resolution settings is part of our roadmap. Our intention is to build a foundation that the community can progressively expand upon, while ensuring that the first version remains tractable and scientifically meaningful.
>
> **(W2)** Thanks for your comment. We fully agree that evaluating against real in-situ observations or reanalysis would be ideal, but the fundamental difficulty is that **no complete historical ocean observations exist** to serve as a ground-truth reference. As a result, directly evaluating AI methods on sparse observations leads to highly biased assessments: a model may fit the few sampled points well yet generalize poorly at the global scale, making its real-world reconstruction skill impossible to verify.
>
> The motivation of OceanVerse is precisely to address this long-standing issue. By constructing a digital-twin evaluation based on realistic MNAR sampling patterns, we provide an indirect but necessary benchmark that allows fair comparison of different AI architectures and hyperparameters under real-world sparsity constraints. This ensures that model development and tuning are grounded in a setting that faithfully reflects observational limitations, even though the underlying field comes from a numerical model.
> We view the digital twin not as a substitute for real-world assimilation skill, but as a prerequisite layer of evaluation that establishes whether a method is robust, data-efficient, and capable of handling the sampling structure of real ocean observations. Ultimately, this enables more reliable downstream application to true observational datasets.
>
> **(Q2)** Dissolved oxygen (DO) is a key variable that reflects both marine ecosystem health and climate change. Under ongoing anthropogenic impacts and global warming, the ocean is experiencing a pronounced deoxygenation crisis. However, due to the extreme sparsity of historical DO observations, large uncertainties remain regarding global deoxygenation trends, variability, and driving mechanisms.
>
> While one could consider variables such as temperature or salinity, their scientific understanding is already mature and their observational coverage is much denser; the sparsity challenge is therefore far less severe than for DO. Conversely, extending the benchmark to trace elements such as Hg or Pb is currently infeasible because such measurements require laboratory analysis and are extremely limited in quantity. Our next step is to include major nutrients (N, P, Si), which are sparser than DO and thus pose greater challenges for AI reconstruction. Overall, focusing on DO allows the benchmark to offer a scientifically meaningful yet technically feasible problem for AI researchers, while remaining extensible to broader biogeochemical variables.

---

### Official Review · Reviewer_xnhc · 2025-11-01

**Soundness:** 2
**Presentation:** 3
**Contribution:** 3
**Rating:** 6
**Confidence:** 3

**Summary:**

The paper introduces OCEANVERSE, a large-scale, AI-ready dataset for reconstructing sparse four-dimensional (space × depth × time) ocean observations. The dataset integrates nearly two million historical ocean profiles (since 1900) and three global numerical simulations (CESM2-omip1, CESM2-omip2, GFDL-ESM4). Using a digital-twin sampling strategy, the authors replicate real-world missing patterns by projecting actual historical sampling locations onto simulated “virtual Earths.” This enables complete evaluation while maintaining realistic sparsity. OCEANVERSE supports benchmarking of diverse ML models for reconstructing dissolved-oxygen fields and related environmental variables at 1° × 1° spatial and annual temporal resolution. Baseline experiments compare classical and recent models (XGBoost, MLP, LSTM, Transformer, GRIN, ImputeFormer, TIDER, OxyGenerator). Results show that OxyGenerator and MLP achieve the best performance, with graph-based architectures handling sparse data most effectively. The dataset also enables systematic tests on different data splits and cross-simulation generalization across virtual Earths

**Strengths:**

1. The work provides a major infrastructure contribution to AI for Earth science by releasing a unified, large-scale, multi-source dataset that directly addresses the scarcity of data.
2. The digital-twin evaluation design is well-justified, allowing reconstruction benchmarking with ground-truth access while preserving realistic sampling sparsity.
3. The dataset’s scale and realism far exceed prior spatiotemporal imputation datasets.
4. Baseline experiments are well-structured and provide informative results illustrating how sparsity challenges model complexity.

**Weaknesses:**

1. Although the digital-twin framework is elegant, the paper stops short of quantitative validation that the simulated sampling truly reproduces real-world MNAR patterns. Statistical comparisons of spatial/temporal sampling distributions would strengthen the realism claim.
2. The resolution (1° × 1°, annual) limits small-scale process studies. The paper acknowledges this but does not demonstrate how the framework could be extended to finer scales or variables.
3. Evaluation remains single-task and single-variable (dissolved oxygen); expanding to multi-variable reconstruction (e.g., nutrients or temperature) would better illustrate generality.
4. While computational challenges (e.g., GRIN OOM) are discussed, runtime and scalability comparisons are not reported.
5. There is little examination of why specific models succeed or fail beyond sparsity effects.

**Questions:**

1. How closely does the digital-twin sampling reproduce real-world spatial and temporal observation densities? Have you validated its fidelity quantitatively?
2. How can OCEANVERSE be extended to higher-resolution or multi-variable reconstruction?
3. Are the performance differences among models statistically significant across runs?

---

> ### Author Response · Authors · 2025-11-23
>
> We sincerely appreciate your thoughtful and detailed review, as well as the positive score. We are grateful for your strong recognition of our key contributions — including the value of releasing a unified large-scale multi-source dataset, the well-justified digital-twin evaluation design, the unprecedented scale and realism of OceanVerse compared with prior datasets, and the informative baseline experiments that illustrate how sparsity challenges model complexity. **Your encouraging feedback is truly meaningful to us**.
>
> Next, we provide point-by-point responses to the concerns and questions you raised.
>
> **(W1, Q1)** Thank you for raising this point. We fully agree that validating the realism of the sampling process is crucial for establishing the credibility of a digital-twin framework. We would like to clarify that in OceanVerse, the digital-twin sampling is not generated from a synthetic distribution; instead, it is directly constructed from the real-world observation coordinates (latitude–longitude–depth–time) collected from all historical observation profiles. As a result, the spatial and temporal sampling patterns—including their strong MNAR characteristics—are inherited exactly from the real observational record rather than approximated or simulated. In this sense, the digital-twin sampling intrinsically reproduces the true sampling sparsity and heterogeneity.
>
> **(W2, Q2)** We appreciate your comment. The current resolution is sufficient for studying global-scale, long-term trends, which are central to understanding ocean ecosystem responses under climate change. This resolution also ensures that AI for Science evaluations remain tractable while maintaining scientific relevance. Regarding multi-variable extension, the raw observations for nutrients (N, P, Si) are already included in the current OceanVerse. To construct reliable benchmarks, we only need to incorporate their corresponding high-quality numerical simulation results. Extending to higher spatial and temporal resolutions will require two components: (1) gridding the observations at finer resolution, and (2) generating or obtaining high-resolution numerical simulation data. Both are part of our planned future work.
>
> **(W3)** Thank you for this helpful suggestion. We agree that extending beyond dissolved oxygen would further demonstrate the generality of the framework. We would like to clarify that the raw observations for nutrients (N, P, Si) have already been collected in the current OceanVerse, and we plan to include them in future iterations once their corresponding reliable numerical simulations are incorporated.
> Regarding temperature, we note that temperature reconstruction is a relatively mature problem with extremely dense observational coverage. From the perspective of advancing AI for Science, we therefore prioritize expanding OceanVerse toward nutrients, which remain scientifically important yet significantly more challenging due to their sparse and heterogeneous observations.

---

> > ### Author Response · Authors · 2025-11-23
> >
> > **(W4)** Thank you for your advice. We report the training time and inference time for all baseline methods as follows:
> > | Model        | GFDL Training (s/epoch) | GFDL Inference (s/year) | omip1 Training (s/epoch) | omip1 Inference (s/year) | omip2 Training (s/epoch) | omip2 Inference (s/year) |
> > |--------------|--------------------------|---------------------------|----------------------------|----------------------------|----------------------------|----------------------------|
> > | XGBoost      | 2.42                     | 7.73                      | 1.66                       | 5.21                       | 1.73                       | 5.61                       |
> > | LSTM         | 66.55                    | 0.55                      | 57.09                      | 0.60                       | 58.85                      | 0.55                       |
> > | MLP          | 13.22                    | 0.34                      | 32.31                      | 0.77                       | 30.13                      | 0.39                       |
> > | Imputer      | 65.10                    | 0.89                      | 19.72                      | 0.61                       | 31.28                      | 0.61                       |
> > | TIDER        | 10.58                    | 0.01                      | 10.53                      | 0.02                       | 11.18                      | 0.02                       |
> > | Transformer  | 47.15                    | 0.35                      | 13.82                      | 0.29                       | 13.94                      | 0.35                       |
> > | Oxygenerator | 354.43                   | 9.10                      | 367.19                     | 9.74                       | 339.11                     | 9.64                       |
> >
> > By the way, we would like to clarify for you and the other reviewers that all the baselines we include are existing methods from prior work, and their training and inference efficiency is not related to the goals of our paper. We report their performance only to show how they run on the benchmark we propose. The objective of this paper is to provide an evaluable 4D ocean-element reconstruction dataset under realistic sparsity.
> >
> > **(W5)** Thank you for this helpful suggestion. We provide an analysis of model performance in Lines 367–414 on page 7. Our findings show that OxyGenerator achieves the best performance because it is specifically designed to capture spatiotemporal dependencies under extremely sparse sampling. In contrast, the MLP, despite its simplicity, attains competitive results by leveraging a large number of trainable parameters. A key reason behind these outcomes is the mismatch between model complexity and the effective amount of available data for existing imputation models, e.g., ImputeFormer, TIDER.
> > This benchmark result highlights a broader implication: designing imputation models for scientific domains requires explicitly addressing the challenges of large-scale and highly sparse observation patterns, rather than relying solely on model complexity. We will make this discussion more explicit in the revised version.
> >
> > **(Q3)** All experiments were conducted with five independent runs, and we report the mean and variance of each model’s performance. We also checked statistical significance, and indeed the differences among models are not significant under all setting. We believe this outcome is itself informative: under highly sparse and heterogeneous real-world ocean observations, the effective signal available for learning is limited, which compresses the performance gap across models. As a benchmark, this highlights the need for developing more robust and data-efficient imputation methods for such challenging scientific scenarios.

---

### Official Review · Reviewer_ypMP · 2025-11-07

**Soundness:** 3
**Presentation:** 2
**Contribution:** 3
**Rating:** 4
**Confidence:** 4

**Summary:**

Basically, the problem is this: imagine you are using a dataset of real ocean observations. If we want to train a neural network on this data, the only way we can validate it is by either partially masking the complete observations or making it predict other observations. Thus, by training only on observations, it is certain that they do not cover the whole ocean, and therefore some dynamics are missed and never learned by the neural network in regions with no data. So, how can you have any confidence in the completely reconstructed states from pure observations in regions where we had no observations? To overcome this issue, they propose: imagine we have access to numerical simulations of the ocean. We are going to subsample them at the locations of the dataset of observations we have. In other words, we create this virtual dataset of observations where all observations are from the numerical simulation but at the very same locations. By doing so, it means that: 1. The state mask is not chosen at random, meaning that the mask allowing us to extract observations has real meaning (MNAR); 2. We have access to a ground truth in regions with no observations to check whether or not the dynamics learned by the neural network work also in these regions. Once we are happy with the trained model, we can apply it to our true observations to reconstruct our states.

**Strengths:**

First of all, I would like to say that I am currently working on building ocean emulators, and very recently I had this debate about how to properly validate our models if we train purely on observations. This paper comes at the right moment! Let me start with the positive aspects:

1. I recognize that the effort required to assemble all the observation databases must have been tremendous. Thank you for that contribution.

2. From a scientific standpoint, I agree that this dataset will surely be very useful for pushing forward the development of new AI emulators based solely on observations. We are moving in this direction for atmospheric emulators, but nothing comparable exists yet on the ocean side, so this will certainly help.

3. I verified that both the codebase and dataset are indeed publicly available. I did not test downloading anything yet, but I saw that everything is already online, so thumbs up.

In conclusion, I really appreciate the effort, and I believe the dataset will be of great use for the scientific community aiming to develop new AI tools for emulating the oceans.

**Weaknesses:**

Nonetheless, I have several concerns with the paper:

1. First of all, let's discuss the scope of the paper. In my view, the paper should focus on presenting your dataset and how you constructed it, etc. As your title indicates, you are presenting a dataset. Therefore, I do not see the point of everything that comes after page 6, starting from the experiments section. I am by no means criticizing your experiment section; however, I think that either the paper should focus exclusively on the dataset or on the creation of the oxygen maps. This is my main criticism.

2. Even considering what I just said, one aspect that I found particularly disappointing in the experiment section is that, despite all the claims made previously about being able to perform robust validation of neural networks trained on virtual Earth and then apply them to true observations, there were no results showing oxygen maps generated from a model trained on true observations...

3. Finally, I am not a fan of the overall writing style of the paper. This is a research article, and for example, the opening sentence is not really appropriate. In addition, I found all the explanations from page 2 to page 6 quite repetitive, and I am pretty sure they could be shortened to half their current length.

**Questions:**

1. **Clarify the paper's focus and scope**: While the dataset construction effort is great and well-executed, the paper suffers from a lack of clear focus. I suggest two possible directions:
   - **Option A**: Reframe the paper exclusively as a dataset contribution, removing or significantly condensing the experiments section. The dataset itself is a valuable contribution that could stand on its own. This would allow for more space to detail the dataset characteristics, validation procedures, and potential use cases.
   - **Option B**: If you wish to present both the dataset and the oxygen reconstruction methodology, the paper needs to be restructured with a clearer narrative that justifies why both contributions belong together !

2. **Improve the writing style and maintain scientific rigor**: This is a technical research paper for ICLR, and the writing should reflect this throughout:
   - The opening sentence of the introduction ("Since hunters from Siberia crossed the Bering Strait to reach Alaska 25,000 years ago...") is too narrative and not appropriate for a machine learning venue. Consider starting directly with the scientific problem.
   - Avoid repetitive statements about how "OceanVerse will create new opportunities" or similar claims. This phrase appears multiple times throughout the paper (particularly in sections 1-3) without adding substantive information. State this claim once clearly, then focus on technical details.
   - Sections 2-6 contain significant redundancy and could be condensed to approximately half their current length without losing essential information. Focus on technical precision over narrative repetition.

3. **Include experiments on real observations**: Given your extensive claims about robust validation through the digital twin approach and the ability to apply models trained on virtual Earth to real observations, it is surprising and disappointing that no results are shown for models trained on real observations. I strongly recommend adding at least one experiment demonstrating:
   - A model trained on your real observation dataset
   - Its performance compared to models trained on the virtual Earth simulations
   - Discussion of the domain gap between simulated and real data
   This would significantly strengthen the paper's practical impact and validate your methodological claims.

---

> ### Author Response · Authors · 2025-11-23
>
> We sincerely thank you for your thoughtful and encouraging review. We truly appreciate your recognition of the substantial effort required to assemble the observational databases, as well as your positive assessment of the scientific value of OceanVerse. As you pointed out, the challenge of validating AI emulators trained purely on observations is both timely and fundamental. In fact, **the motivation for this work grew directly from similar debates with our oceanography collaborators**. Through examining the existing literature, we realized that the irrecoverable nature of historical ocean observations makes fair and comprehensive evaluation extremely challenging. This gap strongly motivated us to develop OceanVerse.
>
> Next, we provide point-by-point responses to the concerns you raised. Many of your comments are extremely helpful for improving our work, and we hope that our clarifications will strengthen your confidence in our paper. We sincerely look forward to receiving your support.
>
> **(W1, Q1)** Thank you for raising this insightful point. Our decision to include the experimental section stems from the diversity of the ICLR readership. As a top conference in machine learning, ICLR attracts both AI for Science researchers—who are interested in how a dataset can support scientific evaluation—and researchers focused on developing new AI methodologies. This dual audience is also reflected in the two branches of related work included in our paper. For this reason, we intend OceanVerse to serve two purposes. First, for AI for Science researchers, we aim to demonstrate a new evaluation perspective for models trained purely on sparse and heterogeneous ocean observations. Second, for researchers developing machine learning methods, we hope to provide a realistic and challenging benchmark—summarized in Table 1—that captures key scientific constraints often overlooked in synthetic or idealized settings. We acknowledge the concern regarding scope and will further clarify the intent in the revised version.
>
> **(W2, Q3)** Thank you for pointing this out — we fully understand your expectation here. We actually debated this issue during the paper preparation. Our initial decision to omit the reconstruction results from models trained on real observations was motivated by concerns about keeping the main paper concise. As you know, presenting and interpreting such results inevitably involves extensive scientific discussions (e.g., the placement of major oxygen minimum zones, long-term trend characteristics, and regional discrepancies), which could easily shift the focus away from introducing the dataset itself.
> That said, we agree that providing these results would strengthen the completeness of the experimental section. We will include the corresponding reconstruction outputs in the appendix of the revised version to better address this concern.
>
> **(W3, Q2)** Thank you for this valuable suggestion. We will revise the writing to make it more concise, remove unnecessary repetition, and ensure that the tone is fully appropriate for a research article.

---

> > ### Comment · Reviewer_ypMP · 2025-11-26
> >
> > Good Morning!
> >
> > Thanks a lot for your replies and for addressing my concerns :-) I’m looking forward to seeing your revision of the paper. If you add your results on true observations (W2, Q3) to the appendix and make a small revision to the paper’s flow (W3, Q2), I’ll be more than willing to change my presentation note (depending on what you’ve done) and also raise my overall score from 4 to 6!
> >
> > Please, I know we’ve asked you for quite a lot, but maybe you could send a preliminary revision by Sunday so that I have a bit of time to thoroughly read your new version of the paper before the incoming deadline.
> >
> > Have a great day.

---

> ### Author Response · Authors · 2025-11-26
>
> Thank you very much for your positive response. Due to the tight rebuttal timeline, we tried our best to make some preliminary revisions to the paper when submitting the rebuttal (the updated version has already been uploaded to OpenReview, and you can view it now). Specifically, on page P19 in the appendix, we added the model’s inference results on the real-world dataset. We plotted the dissolved oxygen distributions at 0 m, 100 m, 500 m, 1000 m, 2000 m, and 5000 m for the year 1980 for illustration. All text modifications are highlighted in red.
>
> We are currently making more detailed and targeted revisions based on your comments. We will upload the fully revised manuscript as soon as possible, and we sincerely hope to receive your support.

---

> ### Author Response · Authors · 2025-11-29
>
> Dear Reviewer,
>
> Due to the unexpected accident, we are sorry that we may not have the opportunity to engage in further discussion with you, but we would still like to remind you of our manuscript update. In addition to adding the model results based on real observations, we have also included in Appendix P20–21 a new comparison of the differences between the model simulations and the observations. We hope this provides a more comprehensive understanding of the data. As you mentioned earlier, our response has addressed your concerns, and we sincerely appreciate your positive evaluation of our rebuttal. We genuinely hope that the new area chair will take note of our discussion, and we very much look forward to receiving the AC’s recognition.

---

### Official Review · Reviewer_VvY3 · 2025-11-08

**Soundness:** 2
**Presentation:** 3
**Contribution:** 2
**Rating:** 4
**Confidence:** 4

**Summary:**

This document introduces OCEANVERSE, the first evaluable 4D ocean element reconstruction dataset, aiming to address the problems of highly sparse historical ocean observation data and the lack of AI-ready data. The dataset integrates nearly 2 million real ocean profile data points with three CMIP6 numerical simulation results, constructing a dataset that conforms to real missing patterns (MNAR) through digital twin sampling. It supports spatiotemporal reconstruction of global marine biogeochemical elements (with dissolved oxygen as a case study). Experiments validate the performance of 8 baseline models, proposing temporal split as the optimal data partitioning method. The dataset provides a standardized benchmark for AI-driven marine science research, with both the dataset and code open-source.

**Strengths:**

1.	It’s the first 4D dataset that realistically handles missing ocean data, which previous AI-ready datasets failed to do.
2.	The article itself is well-organized, easy to follow, and clearly explains the concepts.
3.	OCEANVERSE focuses on pressing scientific issues such as ocean deoxygenation, which is closely linked to climate change and marine ecosystem health. This represents a new, high-potential direction for research in this field.

**Weaknesses:**

1.	Resolution is too low: The data (1°x1° spatial, annual) is too coarse. It's not detailed enough to study smaller or faster ocean processes, like coastal events or seasonal changes.
2.	Unquantified Discrepancy Between Simulation and Real Data: While the document mentions "calibrating virtual Earths with real observations," it fails to quantify the calibration accuracy (e.g., regional or depth-specific deviations between simulated and observed values). Additionally, the potential impact of inherent systematic errors in CMIP6 models (e.g., GFDL-ESM4’s biases in simulating high-latitude ocean processes) on reconstruction results is not analyzed, which risks overestimating the transferability of models trained on OCEANVERSE to real-world ocean scenarios.
3.	Only focuses on oxygen: Despite incorporating multiple environmental variables as inputs, OCEANVERSE centers exclusively on dissolved oxygen as the core target variable, lacking coverage of other ecologically and biogeochemically critical marine elements (e.g., nutrients like phosphate, contaminants like mercury). This narrow focus restricts the dataset’s utility for broader marine science research (e.g., studying nutrient-oxygen coupling or ocean acidification).

**Questions:**

1.	How to quantify the matching accuracy between real observations and virtual Earth data in digital twin sampling? Are there regional adaptability differences?
2.	What is the basis for selecting these other marine environmental variables as inputs?
3.	Is there a clear plan for extending the dataset to other marine elements (e.g., phosphate, mercury or even the mentioned environmental variables)? How to maintain data consistency and unified evaluation standards during expansion?
4.	Are there targeted data augmentation or model optimization suggestions for complex coastal areas and data-sparse regions in the Southern Hemisphere?
5.	What is the spatiotemporal resolution enhancement plan for the dataset? How to balance data sparsity and computational feasibility after enhancement?
6.	What is the impact mechanism of differences between different numerical simulation results on model performance in the generalization evaluation?
7.	The experiments use RMSE and R² as core metrics. Does the dataset's evaluation system lack indicators directly related to ecological effects (e.g., prediction accuracy of deoxygenated area, matching degree with coral reef survival thresholds)? Relying solely on statistical error metrics may disconnect model evaluation from real marine scientific issues, making it impossible to measure the model's supporting value for ecological conservation decisions.

---

> ### Author Response · Authors · 2025-11-23
>
> We sincerely thank you for reviewing our manuscript and for recognizing that OceanVerse represents the first AI-ready dataset reflecting realistic observation gaps, with strong potential for advancing climate change and marine ecosystem research. We also appreciate your positive comments on the clarity and organization of our paper. Below, we provide point-by-point responses to the weaknesses and questions you raised.
>
> **(W1)** Our study targets global, long-term ocean climate variability, including ENSO and PDO, which drive the expansion of oxygen minimum zones (OMZs) at basin to global scales. The current spatial (1°×1°) and temporal (annual) resolution is appropriate for these objectives and aligns with research priorities [1–3]. All three simulation datasets were produced at this resolution, ensuring consistency across sources. While finer resolution could resolve coastal or seasonal processes, our focus is on a globally consistent AI-ready dataset for large-scale oxygen–climate interactions.
>
> [1] A committed fourfold increase in ocean oxygen loss. *Nature Communications* (2021).
>
> [2] Responses of horizontally expanding oceanic oxygen minimum zones to climate change based on observations, *Geophysical Research Letters* (2022)
>
> [3] Assessing the Observational Uncertainties of Dissolved Oxygen Climatology and Seasonal Cycle Through a Coordinated Intercomparison Project, *Global Biogeochemical Cycles* (2025)
>
> **(W2)** We would like to clarify that our manuscript does not mention "calibrating virtual Earths with real observations" after our careful recheck. To address CMIP6 model diversity, we constructed three independent datasets for cross-model comparison and generalizability analysis (Figure 9). While simulation–observation gaps exist due to known model biases, OceanVerse serves as an AI benchmark, allowing assessment of robustness, domain transfer, and uncertainty quantification, rather than providing bias-corrected reanalysis.
>
> **(W3)** We agree that expanding to other key biogeochemical variables would further enhance the dataset’s scientific scope. We started with dissolved oxygen (DO) because it is one of the most extensively observed and scientifically critical variables for studying ocean deoxygenation and climate-driven ecosystem change. In contrast, global observations of variables such as phosphate or mercury are extremely sparse, posing major challenges for deep learning–based reconstruction.
> Nevertheless, we are already collecting and harmonizing datasets of nitrate, phosphate, and silicon, and plan to release an extended version, OceanVerse-Nutrient, to facilitate research on nutrient–oxygen coupling and related biogeochemical processes in the near future.
>
> **(Q1)** In our digital twin sampling, we use only the geographical coordinates (latitude, longitude and depth) of real observations to locate corresponding grid points in numerical simulations, without using the observed values themselves. Therefore, no "difference" in value space exists by design.
>
> **(Q2)** The input variables were determined through discussions with ocean scientists collaborating on this work. They were selected based on the physical and biogeochemical mechanisms governing the changes of oxygen in the ocean.
>
> **(Q3)** Extending the dataset to additional marine elements is indeed planned, with phosphate, nitrate and silicon already collected and harmonized from multiple sources. For the upcoming OceanVerse-Nutrient version, we will adopt consistent data management protocols to ensure unified formatting, quality control, and evaluation standards across all variables.
>
> **(Q4)** This question falls outside the scope of the current study. However, a potential strategy for future work could be to adaptively expand the temporal window in data-sparse regions during reconstruction, allowing models to leverage more contextual information and improve robustness in areas with limited observations.
>
> **(Q5)** Our current plan is to increase the temporal resolution to monthly and simultaneously incorporate additional biogeochemical elements.
>
> **(Q6)** Similar to domain generalization studies, we find that model generalization is directly influenced by differences between numerical simulation methods. For example, reconstructions transfer more easily between CESM2-omip1 and CESM2-omip2, which share more similar physical and biogeochemical representations.
>
> **(Q7)** Currently, the objective evaluation metrics are RMSE and R², which are standard for reconstruction accuracy. The ecological thresholds you mention, such as the precise location of deoxygenated zones or matching with coral reef survival limits, remain highly uncertain and debated within the ocean science community, with no clear consensus. This scientific uncertainty is precisely why collaboration between AI researchers and marine scientists is needed: AI methods can help explore, quantify, and eventually reduce these knowledge gaps in climate ecosystem research.

---

### Official Review · Reviewer_1nZp · 2025-11-09

**Soundness:** 4
**Presentation:** 4
**Contribution:** 4
**Rating:** 8
**Confidence:** 3

**Summary:**

The proposed work introduces a new dataset called OCEANVERSE, which is created using digital twins to simulate real-world ocean observations and handle missing data on a virtual Earth. This dataset includes ground-truth data that ensures robust performance validation. The study uses dissolved oxygen as a case study to demonstrate how the datasets are constructed and also presents baseline models. This dataset is valuable for the scientific community and will undoubtedly advance research on ocean observations (specifically dissolved oxygen) utilizing machine learning.

**Strengths:**

1) **Well-written paper:** The paper is clearly articulated and easy to follow. The graphical illustrations significantly enhance the understanding of the work.

2) **Valuable dataset:** The dataset holds great potential for the scientific community.

3) **Thorough experimentation:** The experimentation section is well-presented, including appropriate baselines and thorough discussions on the results obtained.

**Weaknesses:**

1) **Need for more details:** Additional information is needed regarding the construction of the dataset. What sparse data representation techniques are employed in this work?

2) **Clarification on regression models:** In Figure 3, which regression model is utilized? Besides the model mentioned in the figure, are any other environmental variables used to determine dissolved oxygen? The figure legend (mentioning density, pressure, etc.) suggests that additional variables may be considered.

**Questions:**

Please refer to my comments. I would like to see a few more information about the models and the construction of the dataset

---

> ### Author Response · Authors · 2025-11-23
>
> We sincerely appreciate your positive evaluation of our work, including your recognition of the clarity of our writing, the scientific value of our dataset, and the completeness of our experiments. **Your encouraging comments are very motivating to us, and we hope that, with your support, other reviewers will also recognize the contribution and scientific merit of our paper**.
>
> Regarding the two questions you raised:
>
> - **(W1)** As described in Section 3.2, the dataset was constructed using a digital twin approach. The sampling of sparse observations was achieved by extensively collecting multi-source dissolved oxygen measurements and spatially locating them within numerical simulations based on their geographic coordinates. During the reconstruction process, as our dataset is designed as a resource for general use, we do not restrict it to any specific AI representation method. In our study, we demonstrate eight different approaches for feature representation and ocean element reconstruction.
>
> - **(W2)** Figure 3 presents an overall reconstruction framework utilizing our proposed OceanVerse dataset, which likewise does not depend on any specific regression model. Our dataset provides the same environmental variables shown in the Figure 3, which includes temperature, salinity, nitrate, phosphate, chlorophyll, density, and pressure. All data files are provided in our anonymous resource link for access and verification.

---

### Author Response · Authors · 2025-11-23

# General Response — Positive Feedback Summary

We sincerely thank all reviewers for their constructive feedback and for recognizing the significance and contributions of OceanVerse. Below we summarize the key positive points highlighted by the reviewers:

## 1. Importance and novelty of the contribution
Multiple reviewers emphasized that OCEANVERSE addresses an urgent and long-standing challenge in ocean and climate science:

- Reviewer **QQvM** noted that the work "*addresses an under-explored yet crucial area*" and provides infrastructure that could "*meaningfully advance the emerging field of AI for Oceanography*."
- Reviewer **VvY3** highlighted that this is "*the first 4D dataset that realistically handles missing ocean data, which previous AI-ready datasets failed to do*."

## 2. Realistic and scientifically meaningful sampling design
Reviewers consistently recognized the value of the digital-twin MNAR sampling strategy:

- Reviewer **QQvM** described it as "*well-motivated and technically elegant*."
- Reviewer **xnhc** emphasized that it "*enables complete evaluation while maintaining realistic sparsity*."
- Reviewer **ypMP** explicitly acknowledged that this design addresses a major open problem in validating ocean ML models trained purely on observations —— “*This paper comes at the right moment!*”.

## 3. Value of the dataset to the scientific community
Several reviewers praised the dataset’s potential impact:

- Reviewer **1nZp** stated that the dataset "*will undoubtedly advance research on ocean observations*."
- Reviewer **VvY3** highlighted that it "*focuses on pressing scientific issues such as ocean deoxygenation*."
- Reviewer **ypMP** appreciated the substantial effort in assembling ∼2M observations and concluded that it will be "*of great use for the scientific community*."

## 4. Quality of exposition and completeness of experiments
Reviewers also positively evaluated the clarity and technical execution of the paper:

- Reviewer **1nZp** described it as "*well-written, clearly articulated and easy to follow*."
- Reviewer **xnhc** and **1nZp** both praised the graphical illustrations and the well-structured baseline experiments.
- Reviewer **xnhc** emphasized that the dataset scale and realism "*far exceed prior spatiotemporal imputation datasets*."

We are grateful for these encouraging comments and are pleased that reviewers found OCEANVERSE to be a timely and high-impact contribution.

---

> ### Author Response · Authors · 2025-11-23
>
> # General Response — Common Concerns
>
> Across the reviews, we identified three core thematic concerns raised by multiple reviewers. Below we summarize these concerns and clarify our design choices. In addition, we have provided detailed, point-by-point responses under each reviewer’s chat box.
>
> ## 1. Scope of the paper: dataset-only vs. dataset + experiments
> Reviewers expressed differing expectations regarding whether the manuscript should focus solely on constructing the dataset or should additionally contain reconstruction experiments.
>
> Our intention is deliberate: **OceanVerse is designed to serve two complementary ICLR readerships**.
>
> - **AI for Science researchers**, who need a scientifically grounded benchmark to evaluate reconstruction skill under realistic sparse observations.
> - **ML method researchers**, who need a standardized and challenging benchmark to compare architectures.
>
> The baseline experiments therefore play an essential role:
>
> - They demonstrate how the dataset can be used in practice.
> - They reveal the challenges created by realistic MNAR sampling.
> - They provide a first benchmark for future model development.
>
> At the same time, we avoid including extensive scientific interpretation to keep the focus on the dataset. This design choice balances reviewers’ different expectations while maintaining the paper’s clarity and scope.
>
> ## 2. Choice of dissolved oxygen and the spatiotemporal resolution
>
> Several reviewers questioned why the benchmark centers on DO as the sole reconstruction target, and why we use a 1°×1° annual resolution. These decisions are scientifically and technically motivated:
>
> - Dissolved oxygen is both critical and extremely sparse, unlike temperature or salinity (densely observed) or trace metals (too scarce). DO reflects global climate–ecosystem interactions and remains a major open research topic.
> - The chosen resolution corresponds to the scale of global deoxygenation, ENSO/PDO variability, and Oxygen Minimum Zone (OMZ) expansion. It also ensures tractability and consistency across simulation sources.
>
> Importantly, OceanVerse is designed as a foundation. We have already collected multi-source observations for nutrients (N, P, Si), and plan to include them—along with finer temporal resolution—in the next version once corresponding high-quality simulation fields are processed. This ensures future extensibility while keeping the first release scientifically meaningful and practically usable.
>
> ## 3. Use of digital-twin evaluation instead of real observations or reanalysis
>
> Some reviewers asked whether digital-twin evaluation is sufficient or realistic. The motivation is fundamental:
>
> - There is no complete global historical observation dataset that can serve as ground truth for evaluating AI models
> - Evaluating directly on sparse observations produces highly biased scores and cannot assess global reconstruction capability.
> - Our digital-twin sampling is built from real observation coordinates, faithfully reproducing true MNAR patterns rather than synthetic sparsity.
>
> Thus, the digital-twin provides a necessary and fair evaluation layer for AI models before applying them to real-world sparse datasets.

---

### Meta-Review · Area_Chair_12ZN · 2026-01-07

**Summary:**

This paper introduces a 4D ocean element reconstruction dataset. New, high-quality scientific datasets are indeed valuable for benchmarking AI methods, and there is clear community interest in ocean-focused data resources. However, reviewer sentiment is mixed, with concerns spanning dataset resolution, presentation quality, and the lack of real in-situ or reanalysis observations.

A key issue is that it remains unclear what unique challenges this dataset is intended to capture and how it meaningfully advances the current landscape of ocean benchmarks. In particular, I am not convinced that the dataset, in its current form, can serve as an “ERA5-equivalent” for oceanography. The evaluation and benchmarking setup also appears limited: it is unclear whether the considered metrics are sufficient to capture the relevant scientific and spatio-temporal properties of the reconstruction task, and additional diagnostics may be necessary to demonstrate value for the broader community. The considered baseline models are limited. Moreover, the paper does not clearly articulate a long-term plan for dataset maintenance, versioning, and accessibility, which is an important criterion for dataset/benchmark contributions.

While the authors attempted to address several concerns during rebuttal, many issues remain, and the revised manuscript does not substantially improve clarity or strengthen the case for impact. Given the concerns about dataset design, evaluation, and presentation, I believe the submission would require a major revision to meet the bar expected for a datasets and benchmarks track. I therefore cannot recommend acceptance in its current form.

**Reviewer Concerns:**

The authors used the rebuttal phase to address many of the reviewers’ concerns. However, fundamental questions remain regarding the dataset resolution, the strength and completeness of the baseline evaluation, and the reliance on digital-twin evaluation rather than validation against real observations or reanalysis data.

**Reviewer Scores:**

I do not expect the discussion phase would have shifted the scores substantially.

---

### Decision · Program_Chairs · 2026-01-26

Reject